# Dynamic adversarial neural cryptography for ensuring privacy in smart contracts

Basil Hanafi[1], Mohammad Ubaidullah Bokhari[2], Mudasir Ahmad Wani[3], Kashish Ara Shakil[4] and Gauhar Ali[3]

[1] Department of Computer Science, Aligarh Muslim University, Uttar Pradesh, India
[2] Department of Computer Science, Aligarh Muslim University, Aligarh, UP, India
[3] EIAS Lab, College of Computer & Information Science (CCIS), Prince Sultan University, Riyadh, Saudi Arabia
[4] Department of Computer Sciences, College of Computer & Information Science (CCIS), Princess Noura Bint Abdulrehman University, Riyadh, Saudi Arabia



Corresponding author
Mudasir Ahmad Wani,
mwani@psu.edu.sa

## ABSTRACT

Various types of research are being carried out to advance in the field of cryptography and develop a more robust technique for security. Adversarial neural cryptography (ANC) is a recent development in this extension, which possesses huge potential to be implemented in various domains. There is a continuous need for the development of more adaptive techniques to secure data while in communication using deep learning and other applied artificial intelligence techniques, which serves as the motivation for this work stems from the increasing need for adaptive, robust encryption mechanisms to address the limitations of traditional cryptographic techniques in securing sensitive blockchain transactions. This article proposes a new approach for the protection of private smart contracts on blockchain systems *via* ANC. The proposed method in the research is dynamic adversarial training using three neural networks to secure smart contract transactions. It optimizes the encryption and decryption processes against evolving cyber threats. The algorithm strives to attain a high key agreement rate (KAR) and a lower Eve's decryption failure rate (EDFR) eventually proving its efficacy in attaining privacy and security in blockchain applications and adaptability. This research will incite more studies on ANC and its practical implementations in ensuring private smart contracts and overcoming the present cryptographical approaches with significant development because of their potential.

## INTRODUCTION

The increased dependence on decentralized systems and blockchain (*Zhao, 2019*) has led to transaction and agreement execution fundamentally changing how it all happens in terms of safety, transparency, and immutability. In fact, even though robust infrastructure exists on blockchain platforms like Ethereum for smart contracts, they face some serious issues relating to privacy and security. Traditional public blockchain networks expose all this transaction data, which is a risk when sensitive information is involved. On a more practical note, as more sectors of such industries as finance, healthcare, and supply chain

management begin to adopt blockchain technology (*Wang et al., 2019*), it becomes an absolute necessity to protect the confidentiality of such transactions.

Smart contracts are self-executing contracts written with the terms of an agreement directly into code. One of the critical components of blockchain is that it ensures both trust and automation in a decentralized environment, and within the very same structure lies an inherent transparency that may expose critical data (*Sharma, Jindal & Borah, 2023*). So, possibility-wise, their vulnerability to unauthorized access or attacks is a possibility. These problems were solved by private smart contracts that allowed sensitive transactions to be carried out in a more controlled and confidential environment. Although several cryptographic methods have already been adapted to secure smart contracts, such as symmetric and asymmetric encryption, the increasing complexity of cyber threats has motivated research into advanced and adaptive approaches. Among these, adversarial neural cryptography (ANC) has emerged as a promising direction; however, it remains an exploratory technique without established security guarantees, particularly when compared to more rigorously analyzed post-quantum primitives such as lattice-based cryptography.

In the context of this research, on the side of the development of blockchains, there lies ANC, an emerging field that constructs neural networks and cryptography into each other (*Abadi & Andersen, 2016*). Hence, here it is a promising approach to the improvement of the security of private smart contracts (*Kumar et al., 2024*). ANC employs adversarial training between neural networks-there are usually three entities namely encoder, decoder, and adversary-to dynamically evolve encryption methods able to resist sophisticated attacks. This approach brings about a self-adaptive framework that continues to enhance its capabilities in protecting communications (*Coutinho et al., 2018*). Potentially, this shows a solution to the limitations of traditional cryptographic techniques.

This article proposes adding the concept of ANC to secure private smart contracts on blockchain platforms. The main contribution of this article would be the development of an algorithm that applies ANC for improvement in smart contract transaction executions as having more desirable privacy and security properties than traditional encryption methods by offering resistance to advanced cyber threats through a dynamic adversarial training methodology involving multiple neural networks.

The major contributors to the security of smart contracts include traditional cryptographic techniques, including symmetric encryption, homomorphic encryption, and zero-knowledge proofs. However, these methods suffer from several inherent drawbacks such as static encryption mechanisms and high computational costs that limit their adaptability to emerging cyber threats. This research aims at building on the foundation developed in ANC, by constructing a dynamic adversarial training framework for smart contract security. In contrast to basic ANC, which is often oriented towards generalized secure communications, the developed method would specifically introduce an algorithm to the domain of secure private contracts within blockchain environments. Enhancements consist of optimized adversarial training frequencies, noise resilience integrated to prevent the adversary Eve from decrypting data, and a structured test framework that can achieve a 98% key agreement rate (KAR). These advances bring

forward the proposed approach, as it is not only computationally efficient with 10% overhead compared with existing cryptographic systems like Advanced Encryption Standard (AES) and Rivest–Shamir–Adleman (RSA) but also is significantly more robust against an adversary.

The market for smart contracts is exponentially growing, having already exceeded the value of $300 billion (*George, 2024*) at the beginning of the previous year. However, such growth has opened massive security risks. In the first half of 2023, hackers stole nearly $750 million using vulnerabilities in smart contracts. These contracts often depend on external data, and audits have shown that on average, 117 external data accesses occur per project, while some have even reached as high as 11,724. Such extensive dependence increases the possibility of security breaches. In addition, the complexity of building smart contracts correctly means that even minor coding errors can lead to severe consequences, such as unauthorized access and substantial financial losses (*Scharfman, 2024*). Historical incidents like the DAO hack in 2016 and theft through Parity Wallet in 2017 remind us of the existing costly flaws in today's smart contract systems.

The growing sophistication of cyber-attacks against blockchain systems, and the sensitive nature of the transactions within smart contracts, highlight the need for cryptographic solutions that are adaptive and resilient. Existing ANC frameworks, even though effective for generic tasks associated with encryption, do not provide particularly specific optimizations for blockchain use cases in terms of security, especially in maneuvering private smart contracts. The motivation to create this work stems from filling in these gaps, as the ANC model would be designed specifically with the blockchain's specific needs in mind, including dynamic adversarial attacks and transaction privacy. The proposed model doesn't only incorporate adversarial training but also focuses on practical issues, such as ensuring that smart contracts cannot be changed during adversarial retraining cycles. This work extends the traditional ANC methodology to a self-adaptive encryption framework by introducing such innovations as periodic weight updates and noise augmentation so that the system is able to handle real-time and dynamic threat landscapes in decentralized applications. The following are the main contributions of this Research:

- *Dynamic encryption framework:* The article presents a novel dynamic adversarial neural cryptography model that is developed to improve the security and privacy aspects of private smart contracts on blockchain networks.
- *Innovative algorithm design:* This article proposes a customized algorithm where adversarial training among three types of neural networks—encoder Alice, decoder Bob, and adversary Eve—are optimized to establish strong encryption with cyber threats that are sophisticated in nature.
- *Enhanced resilience:* The research model demonstrates considerable resilience in withstanding adversarial attacks and thus counteracts the notion of adaptability over static conventional cryptography methods.
- *Adaptability and scalability:* The dynamic adversarial training mechanism makes sure that the model will learn to adapt to changes in the landscape of cyber-attacks while at

the same time being scalable enough for practical use in decentralized systems with a high Value of Key Agreement Rate.

- **Comparative evaluation and practical insights:** This Research presents a detailed comparison with conventional cryptographic methods being used for Smart Contracts over the trade-off between computational overhead and security, and provides some practical guidelines on ANC adaptability in the real-time environment for blockchain.

This article is further organised as follows: The Introduction section offers an overview of the research and its objectives. The Related Works section discusses prior studies on smart contract security and highlights the significance of privacy in blockchain technology. The Materials and Methods section details the proposed ANC approach and its application in securing private smart contracts. The Results section presents the performance evaluation of the proposed algorithm using key metrics such as the key agreement rate (KAR) and Eve's decryption failure rate (EDFR). The Discussion section provides insights into the obtained results and includes a comparative analysis with existing research. Finally, the Conclusion section summarizes the study's findings, addresses current challenges and limitations, and outlines potential directions for future research in integrating ANC with blockchain technology.

## SMART CONTRACTS AND THEIR USAGE IN EMERGING DIGITAL ECOSYSTEMS

Smart contracts are among the key innovations of blockchain technologies (*Allam, 2018*). A smart contract may be defined as an autonomous code snippet incorporating the terms, conditions, and obligations existing in a contract and executed on a distributed ledger. Smart contracts are distinct from traditional contracts as they execute automatically once pre-specified conditions have been satisfied. The deployment of a smart contract goes through verification and is inscribed on a distributed network of nodes, thereby ensuring consensus, transparency, and unalter ability.

Technically, smart contracts typically consist of functions, state variables, and events programmed with domain-specific languages such as Solidity (Ethereum) or Vyper (*Kaleem, Mavridou & Laszka, 2020*). Once deployed, the contracts can no longer be altered at will due to the nature of immutability of blockchain technology. Although this attribute improves reliability, it also creates some level of rigidity, making upgrades and remedying vulnerabilities more complex. The immutable nature of smart contracts ensures contractual terms are not tampered with; however, this also means that any inherent flaws or security weaknesses are left intact once the contract is posted on-chain. The flexibility of smart contracts has led to their use in a wide range of industries:

- Finance and decentralized finance (DeFi): At the heart of decentralized exchanges, automated market makers, lending protocols, and investment products, including systematic investment plans (SIPs), are smart contracts. By eliminating or minimizing intermediaries, these contracts lower costs, facilitate instantaneous settlement, and expand access to financial products (*Schär, 2021*; *John, Kogan & Saleh, 2023*).

- Supply chain management: Payments are made automatically with confirmation of delivery, quality validation is enforced through Internet of Things (IoT) sensors, and tamper-proof records of goods movement are generated. It is done in a more transparent, less fraudulent, and quicker logistics processes (*Bottoni et al., 2020*).
- Insurance and healthcare: They are utilized to handle safe patient data transfer, automate claims processing, and authenticate clinical trial processes. Blockchain transparency and automation together can cut down on inefficiencies in healthcare systems (*Omar et al., 2021*).
- IoT and energy uses: Decentralized trading of energy using smart contracts enables peer-to-peer selling of electricity, prices are automated, and microtransactions are paid between devices (*Aloqaily et al., 2020*; *Moudoud, Cherkaoui & Khoukhi, 2022*).
- Intellectual property and digital identity: Smart contracts facilitate secure control of digital identities, authentication credentials, and intellectual property rights including licensing arrangements and royalty payments (*Ferro et al., 2023*).

Such applications prove that smart contracts are not specific to financial transactions but are a general-purpose tool for automating trust across disparate digital environments. Though they have the potential to transform, smart contracts come with daunting challenges of paramount importance that directly propagate the urgency for more powerful cryptographic mechanisms.

1. All contract code and state variables are exposed to all nodes in a public blockchain like Ethereum. Transparency is ensured but confidentiality is sacrificed. Sensitive data (*e.g.*, financial decisions, health records, business rules) can leak or be reverse-engineered.
2. Even if direct data exposure is avoided, malicious actors can infer private data from transaction traces, contract outputs, or event logs. For example, finance contracts may reveal investor actions, and supply chain contracts may reveal competitive strategies.
3. Deployed contracts are immutable, so that once a vulnerability or outdated design is found, it cannot be fixed without deploying a completely new contract. This may be a benefit in establishing trust, but rigidity makes adaptive security especially difficult.
4. Solutions like homomorphic encryption, secure multi-party computation, or zero-knowledge proofs give good guarantees but are computationally intensive, so their practical application in throughput-constrained blockchain settings is limited.
5. When blockchain networks are large, it can result in delays and increased transaction costs (*e.g.*, Ethereum gas) to perform sophisticated smart contract logic or cryptographic computations.

Under these constraints, protecting confidentiality in smart contracts needs to be revolutionary with respect to striking a balance between security and efficiency. ANC (*Hanafi, Bokhari & Khan, 2024*) offers an innovative solution by making it possible for models to learn encryption–decryption protocols dynamically under the guidance of an adaptive adversary. In contrast to static cryptographic building blocks, ANC improves over

time through the training of Alice and Bob (communicating parties) against Eve (the attacker), creating adaptive, task-specific encryption that changes with the threat.

In this work, smart contracts are framed as the underlying motivation for deploying ANC. By protecting the confidentiality of contract information and execution logic, ANC solves the transparency–confidentiality dilemma implicit in blockchain. Additionally, the use of noise-hardened adversarial training as well as light-weight fully connected architectures guarantees that the solution is computationally tractable with only small overhead. Notably, to balance immutability with adaptability, we introduce a blockchain-compatible deployment model, where encryption models are retrained off-chain and reintroduced in version-controlled updates without altering the underlying contract logic. Smart contracts augment the strengths of blockchain technology by facilitating automation, trust minimization, and transparency in a wide range of industries. Their effectiveness is, however, challenged by confidentiality risks, immutability problems, and computational compromises that are a characteristic of classical cryptography methods. With the addition of adversarial neural cryptography, this study contributes a dynamic, adaptive, and powerful means of protecting smart contracts from adaptive adversarial attacks. This positions smart contracts not only in a core application field but also at the forefront as a driving force for neural cryptographic research to advance.

## CONTRIBUTIONS AND CHALLENGES ADDRESSED

Smart contract confidentiality in blockchain settings needs adaptive and efficient cryptographic mechanisms. This research makes progress in ANC through the introduction of a learning-based encryption-decryption model that is responsive to adversarial activity with an emphasis on practical deployability. The main contributions of this research are as follows:

1. This research aims to design and implement an end-to-end ANC framework in which Alice and Bob collaboratively learn an encryption–decryption mechanism, while Eve acts as an adaptive adversary attempting to compromise communication. The architecture is specially designed for 32-bit smart contract payloads and encryption keys, thus showcasing flexibility in adapting to blockchain applications where transaction-level privacy is required. The architecture is specifically optimized for 32-bit smart contract payloads and encryption keys, thereby demonstrating versatility in conforming to blockchain use cases where privacy at the transaction level is needed.

2. Adversarial learning with three agents is notoriously unstable and prone to collapse. To overcome this, the proposed framework introduces two stabilization mechanisms: (i) an asymmetric training cadence in which Alice–Bob is updated more frequently than Eve, and (ii) noise-hardened ciphertexts generated through Gaussian perturbations. These controls guarantee consistent convergence, fend off Eve's takeover during training, and enhance the resilience of the learned cryptographic scheme.

3. Experimental evidence reveals that the proposed model has a high rate of key agreement between Bob and Alice, yet, simultaneously, a high rate of decryption failure for Eve. Significantly, these gains are with relatively low extra computational

overhead than in classical symmetric baselines, thereby compromising robust security guarantees with actual practicality in blockchain applications.

4. A distributional analysis of the encrypted outputs shows nearly uniform ciphertext histograms, a characteristic which minimizes structural leakage and maximizes resilience against statistical exploitation. This demonstrates the system to approximate semantic-style security above and beyond traditional reconstruction error metrics.

5. Given the immutability of deployed smart contracts, we suggest a deployment path where the encryption component gets retrained off-chain and re-deployed in a versioned fashion. This allows the privacy layer to adapt against new adversarial attacks while maintaining the immutability of the core business logic and state.

The above was accomplished through systematically solving a number of technical issues that have long held back the development of ANC in decentralized settings. In particular, the framework alleviates adversarial training instability by scheduling and injecting noise, minimizes ciphertext leakage threats through distributional hardening, ensures a best balance of adversarial power by adapting Eve's capacity and learning rate, aligns immutability requirements with flexibility through retraining mechanisms, and promotes computational efficiency using light-weight fully connected networks. This study shows that adversarial neural cryptography can be made viable for securing smart contracts with robust confidentiality, adaptability, and efficiency. Through the synthesis of technical innovation and empirical proof, it adds to the larger literature on neural cryptography and lays the ground for eventual integration into blockchain systems.

## RELATED WORKS

In recent years, research on advanced technologies for securing smart contracts has been an area of great interest with continuous developments. Researchers around the world are working to develop methodologies for enhancing the privacy, security, and reliability of private smart contracts for robust implementation, since smart contracts are significantly important in the broader domain of blockchain applications and adaptability. Numerous innovative cryptographic techniques and algorithms were developed and researched, ranging from zero-knowledge proofs to homomorphic encryption and multi-party computation, which are implemented to tackle inherent vulnerabilities among smart contracts, enhancing their Applicability. The initiatives intend to eliminate such risks, such as unauthorized access and breaches, data, and systemic flaws, that can lead to compromises in blockchain-based systems.

In this context, the research primarily aims to identify and fill the existing gaps in securing smart contracts, which is critical to developing improvements for extending this domain. This research will be positioned towards contributing by taking advantage of ANC as a new approach for private smart contract security. The research seeks to develop a robust framework through the systematic analysis of existing methodologies and their limitations by overcoming the challenges posed by traditional cryptographic solutions. The exploration of adaptive and resilient encryption mechanisms, which highlights the

potential of ANC to address the evolution of cyber threats, will ensure the secure execution of smart contracts in decentralized environments.

*Pranto et al. (2021)* investigate blockchain, IoT, and smart contract integration for smart agriculture with a focus on pre-harvest and post-harvest operations. Their framework utilizes IoT sensors for environmental parameter monitoring in real-time and Ethereum smart contracts for automating traceability, pricing, and distribution. Although the research shows novelty through an event-driven logging working prototype and gas cost analysis, the study's simulation over actual deployment restricts practical generalizability. The authors also recognize issues like excessive gas fees, sensor attack vulnerabilities, and immutability of data as challenges. Areas for future work are deployment on permissioned blockchains, artificial intelligence (AI)-based anomaly detection, and integration with consumer transparency tools, making their research a basis for blockchain-based agricultural traceability.

*Wang et al. (2024)* introduce Ext-ttg, a deep learning framework aiming to solve the issue of multi-label smart contract vulnerability detection complexity. By combining Transformer–bidirectional gated recurrent unit (Transformer-BiGRU) models with extractive summarization, their method optimally handles long opcode sequences and demonstrates higher detection performance than traditional models. The system proved to have a micro-F1 of 82.48% and lower Hamming Loss, confirming its technical stability. Nevertheless, scalability in real-time applications, unexplainability, and specificity to Ethereum smart contracts continue to be major limitations. The research proposes future work in cross-chain data sets, explainable AI paradigms, and lightweight architectures, making Ext-ttg an innovator but improving solution to blockchain security.

*Alkhazi & Alipour (2023)* propose a multi-objective optimization framework for testing smart contracts with concerns for cost, efficiency, and reliability in blockchain applications. Employing NSGA-II, their approach strikes a balance between code coverage, gas expenditure, and execution time, exhibiting better results across case studies of major platforms such as Chainlink and PancakeSwap. The work sets the value proposition of Pareto-optimal solutions in lowering redundant tests while preserving fault detection ability. Its validity is, however, limited to Solidity and Ethereum-based contracts, thus limiting broader applicability to other platforms. Revisions like reinforcement learning–based dynamic optimization, cross-platform verification, and augmented fusion with runtime monitoring would enhance applicability. The research contributes to testing practices by transcending mono-objective approaches towards overall optimization.

*Swetha & Prathap (2025)* interest lies in secure sharing of big data using an Ethereum-based smart contract system that integrates a dilated weighted recurrent neural network (DW-RNN) with a hybrid crypto system. By interweaving elliptic curve cryptography with ElGamal encryption and maximizing key management *via* MBERSO, the architecture provides better authentication, accuracy, and confidentiality than traditional models. Though with promising findings, the work is bounded by use of simulated healthcare datasets and no validation across heterogeneous, real-world settings. Scalability within high-throughput blockchain settings and also in the absence of adversarial testing highly limit practical application. Extensions for future work are

federated learning to ensure privacy-preservation, explainable AI (XAI) incorporation for interpretability, and adherence to regulatory standards like General Data Protection Regulation (GDPR). Their research points to an important step forward in integrating blockchain with AI for trusted data management.

*Munaganuri, Yamarthi & Bolem (2025)* introduce an integrated smart agriculture model that combines IoT soil moisture sensing with predictive analytics, blockchain, and reinforcement learning. Their method showed measurable benefits, such as a 20% decrease in water consumption and a 12% increase in crop yield in field tests, highlighting its potential for sustainable irrigation. Utilization of Hyperledger Fabric for immovable data storage and smart contracts to automate increases scalability and transparency. Nonetheless, the validation of the framework was confined to maize plants and particular environments, which is a concern regarding its generalizability. Computational power, energy expenditure, and adoption limitations also limit deployment in resource-poor settings. Future enhancement could be through lightweight AI models, fusion with satellite and drone data, and hybrid blockchain techniques for enhancing scalability. This contribution is notable for its holistic integration of blockchain, IoT, and AI, though further work is needed to broaden generalizability and enhance socio-technical adoption.

*Kosba et al. (2016)* performed a scientometric analysis of IoT-based air quality monitoring using publication trends, eminent authors, institutions, and keyword networks to identify emerging themes in the field. Their results point to the accelerated development of low-cost sensors, AI calibration, and networked monitoring systems, connecting these to wider smart city and sustainability agendas. Yet the research is founded primarily on quantitative bibliometric measures and a single database, excluding regional contributions and limiting qualitative observations. There is a need for future research incorporating multiple databases, empirical tests, and cross-disciplinary approaches for improving practical relevance.

*Yuan et al. (2018)*, investigated IoT-based frameworks coupled with machine learning for environment monitoring, which is air pollution measurement. The research illustrates how computational models integrated with data obtained from sensors enhance prediction accuracy and responsiveness of the system. Notwithstanding its contribution, the research is based on testbeds at a limited scale and does not consider device interoperability, power consumption, and security of data. Future studies may emphasize large-scale validations, low-power edge computing, and multi-source data fusion to support enhanced predictive performance and sustainable deployment.

*Yuan et al. (2018)*, used machine learning models to forecast urban air quality based on environmental sensor data, presenting regression and ensemble algorithms that performed better compared to statistical baselines. Their experimental setup prioritizes real-time applicability, which is of particular importance to smart city planning. However, the data set is geographically limited, lowering generalizability, and insufficient focus on noisy real-world data and explainability of the model. Larger data sets across regions, using cutting-edge temporal models such as long short-term memory (LSTM), and incorporating explainable AI would enhance robustness and policy relevance.

*Solomon, Weber & Almashaqbeh (2023)* applied deep learning to medical image analysis with an emphasis on the detection of cancer using CNNs with transfer learning. Their tests had greater diagnostic performance than the typical models and showed the utility of transfer learning in managing limited medical data. Nevertheless, the diversity of the dataset used in the study is low and thus its generalizability is questionable, while issues concerning interpretability and computational cost are not addressed. Future studies must involve multi-center datasets, explainable AI techniques, and light CNN models optimized for clinical applications so as to enhance translational value in healthcare.

As *Qi et al. (2024)* systematically reviews privacy-preserving smart contracts (PPSCs), focusing on two primary approaches: cryptographic tools and trusted execution environments (TEEs). They classify the most advanced approaches into two categories, namely crypto-based and TEE-based PPSCs, giving a comprehensive analysis of each scheme's strengths, weaknesses, and challenges. The cryptographic solutions involved include zero-knowledge proofs and multi-party computation. These are known to have strong privacy guarantees but raise many issues concerning the efficiency and scalability of such solutions. TEE-based approaches, such as Intel SGX, provide better performance yet are always associated with hardware trust assumptions. The article points out the major challenges faced in implementing a PPSC, such as preserving privacy during multi-party transitions and enabling efficient generation of proofs, and it furnishes some directions on how to make the system more private, scalable, and user-friendly for future research.

*Jiang et al. (2024)* propose a new type of privacy-preserving smart contract named Regulatable Privacy-Preserving Smart Contracts, particularly suited to account-based blockchains. The proposed scheme is provided in the form of a two-layer commitment structure, in which the detailed fine-grained privacy protection is accomplished with the flexibility for selective disclosure of private data and flexibility for states of transition from private to public data and *vice versa*. The authors also integrate a zero-knowledge proof system with public-key encryption, enabling the system to maintain privacy, soundness, and traceability, while allowing regulators to reveal private data under specific conditions. The system's performance was evaluated on applications such as blind auctions and electronic voting, demonstrating its practicality and effectiveness in real-world blockchain implementations.

*Kosba et al. (2016)* proposed a cryptographic framework to achieve privacy and fairness in blockchain transactions with special attention to the auction sealed-bid case. Here, concealing user transactions while ensuring fairness over the smart contracts is considered. Zero-knowledge proofs and others ensure correctness in the outcome of the transaction without leaking sensitive information. This system proved safe against a variety of adversarial models and showed its efficiency in reducing transaction costs and computational overhead in blockchain networks.

*Aslam, Tošić & Mrissa (2021)* offers a detailed analysis of the privacy and security challenges in blockchain systems with proper justification for secure and privacy-aware design. Some of the key requirements are protecting transaction data, protecting user identity, and other security-related properties such as confidentiality, integrity, and availability. This article describes existing mechanisms, including ring signature,

zero-knowledge proof, and homomorphic encryption, with a comprehensive description of benefits and limitations. In addition to these classifications of different blockchain attacks with countermeasures, the recommendations for designing privacy-preserving blockchain systems are set out. The authors have charted out the future research track toward solving scalability, efficiency, and privacy issues in current blockchain technologies.

*Li et al. (2023)* propose Nereus, an anonymous and secure ride-hailing service including private smart contracts and Software Guard Extensions (SGX)-based technology to boost privacy and security within such services. Among some of the significant challenges that the study attempts to overcome lies the fact of collusion attacks between ride-hailing service providers (RHSP) and drivers that may breach the anonymity of riders. Nereus brings a mechanism for privacy-preserving matching using Bloom filters and Merkle proofs to ensure an efficient and verifiable matching procedure for users. Anonymity is maintained through short group signatures in anonymous authentication and also commitments for deposits enabling accountability for misbehavior on the part of drivers. The authors demonstrate the utility of Nereus with a prototype implementation on the Ethereum network, coupled with an extensive performance analysis clearly demarcated to show computational efficiency over previous systems.

*Yuan et al. (2018)* present ShadowEth, a system envisioned to provide privacy preservation for smart contracts executing on public blockchains, such as in the case of Ethereum, using hardware enclaves. TEEs combined with public blockchain verification preserve the confidentiality of smart contract executions, ensuring integrity and availability, as seen in the case of ShadowEth, where execution of private smart contracts offloads in secure off-chain environments, while all on-chain verification processes take place. They show several use cases demonstrating the applicability of ShadowEth, namely private voting, transaction protection, and auctions. They implemented their system using Intel SGX and provided several security analyses to demonstrate common attacks that the system resisted.

Apart from symmetric and asymmetric cryptographic techniques, other techniques are also being maneuvered for the cause. *Sánchez (2018)* present Raziel, multi-party computation (MPC)-based system wherein Proof-Carrying Code (PCC) is incorporated to enable private and verifiable smart contracts on blockchains. The same framework offers concomitant privacy and correctness by keeping contract information secret while facing the same kinds of attacks as seen in the DAO and Gyges attacks. Here, a new application of non-interactive zero-knowledge proofs is presented to validate smart contracts to third parties without disclosing contract details. They also introduce an incentive scheme for pre-computation of data by miners so that some computations on that data may be rendered safely further optimizing the efficiency of the system. Real-world feasibility is shown through examples like private crowdfunding and double auctions based on Raziel.

*Avizheh, Nabi & Safavi-Naini (2024)* provides an approach for outsourcing verifiable computation, called Refereed Delegation of Computation (RDoC), based on the use of smart contracts. It allows weak clients to delegate their own computation to some untrusted servers while ensuring the correctness of the obtained result, thanks to replication and verification by the third-party smart contract acting like a referee.

They propose security models within the universal composability (UC) framework, addressing challenges in copy attacks where malicious servers' duplicate results obtained from honest servers. Both protocols are intended for two-server and multi-server scenarios, which include formal security proofs. The authors provide an efficiency analysis and an implementation *via* Ethereum to show the feasibility of the system for decentralized computation.

*Zhang et al. (2024)* introduce a blockchain-based proxy-oriented data integrity checking mechanism in cloud-assisted Intelligent Transportation Systems (ITS). The data integrity checking mechanism is an important aspect of data integrity for the storage in the cloud, especially traffic control information, where tampering with the same or making any alteration could lead to extreme consequences. Further, in this mechanism, the computation overhead of the data manager is saved as the task of integrity-checking may be outsourced to a proxy. This ensures complete transparency within the system because a process cannot accommodate malicious behaviors detrimental to its integrity. The proposed mechanism is based on homomorphic hash functions and polynomial encapsulation techniques and this improves efficacy and security toward effective performance in real-time ITS environments.

Research is also being carried out to establish a base of the current scenario for the developments carried out so far in this direction of research. *Hewa et al. (2021)* have performed an extensive survey on blockchain-based smart contracts, with an emphasis on technical matters and future research areas. The article has focused on decentralization, immutability, and cryptographic security of blockchain as key features. This article discusses some of the critical issues regarding the security, privacy, cost of gas, and concurrency of smart contract programming languages such as Ethereum and Hyperledger Fabric. Besides that, it also elaborates on common vulnerabilities of security, such as reentrancy attacks, underflow/overflow errors, and double-spending attacks. Lastly, the article concludes by demonstrating possible future research areas that will integrate smart contracts with artificial intelligence and game theory to make them even more powerful and scalable.

*Steffen et al. (2022)* introduce a ZeeStar-anonymity-preserving smart contract system designed to let developers write private smart contracts without deep cryptographic expertise. It does this by combining the existing zkay language with Non-Interactive Zero-Knowledge proofs (NIZK) and homomorphic encryption. ZeeStar can thereby operate on foreign data that previous systems could not operate efficiently on. The system supports applications including private payments, oblivious transfers, and token transfers on the Ethereum blockchain. With regard to the evaluation of the system, it is practical with a feasible gas cost and has scalability across various smart contract applications.

*Alupotha & Boyen (2022)* propose a novel zero-knowledge protocol for smart contracts called Confidential Integer Processing (CIP), which aims to be used for the sake of private multi-party computations, namely, confidentiality-preserving transactions. The proposed protocols of CIP, namely CIP-DLP and CIP-SIS will offer not only interparty but also intraparty privacy with no dependency on trusted hardware. These protocols provide efficient zero-knowledge proofs for operations like addition, multiplication, and

comparison, and they are secured under the UC framework. The authors demonstrate that their approach improves privacy and security for blockchain applications, such as escrow mechanisms and federated learning, by maintaining confidentiality while still enabling verifiable computations.

*Solomon, Weber & Almashaqbeh (2023)* propose smartFHE-a new framework for privacy-preserving smart contracts with data under encryption using Fully Homomorphic Encryption (FHE). It is architected to enable computations on private data encrypted on such without putting burdens on end users. Instead, the computations are carried out in the blockchain by miners on encrypted inputs and concurrently verified for correctness based on zero-knowledge proofs (ZKPs). The authors discuss challenges that FHE poses towards efficiency and concurrency in blockchain environments and prove how smartFHE can be applied for various use cases like private payments and decentralized applications. Performance evaluations prove that smartFHE is scalable and efficient enough to be used by lightweight users in blockchain settings.

To address the usage of zero-knowledge proofs, *Lavin et al. (2024)* present a sweeping survey of the use cases of ZKPs, which puts more emphasis on the flexibility of privacy and the computational integrity of the most diverse application domains. There, the authors explain how ZKPs, specifically zk-SNARKs, quietly incorporated blockchain technologies to boost privacy, scalability, and interoperability. Further, they also discuss non-blockchain applications like voting systems, authentication, and machine learning. The article concentrated on foundational components of the ZKP infrastructure, with a focus on zero-knowledge virtual machines (zkVMs) and domain-specific languages (DSLs). The article focuses on the growing importance of ZKPs in future cryptographic practices and digital privacy, such that they represent an essential avenue for developing secure, privacy-preserving technologies.

Hence, all the above-mentioned research in this section can be applied to achieve privacy-preserving smart contracts, confidential computing for smart contracts, smart contracts with homomorphic encryption, multi-party computation, and attribute-based encryption. For a quick contrast comparative analysis of all the cryptographic techniques being utilized can be observed in Table 1.

As all the techniques can be analyzed based on various parameters in this section, after a confined tabular comparison, giving a spectrum of applicational research advancements to secure private smart contracts. This section provides an in-depth review of existing methods currently available for securing private smart contracts. Though the approaches discussed here are useful, several research gaps can be identified that served as a base for this research are listed below:

(1) Zero-knowledge proofs and homomorphic encryption are computationally expensive cryptographic methods, which limit their applicability in high-throughput blockchain environments. In parallel, TEE-based solutions, though efficient, rely on specialized hardware, which restricts their scalability and accessibility in decentralized systems.

(2) Traditional cryptographic and TEE-based techniques are not responsive to new cyber threats that dynamically evolve. It's a static nature that makes a sophisticated attack inevitable in fast-moving environments.

**Table 1  Cryptographic techniques for securing private smart contract.**

| Technique | Privacy level | Performance | Scalability | Complexity | Security | Cost | Compatibility | Standards | Maturity | Use cases |
|---|---|---|---|---|---|---|---|---|---|---|
| Homomorphic encryption | High | Low | Medium | High | High | High | Medium | Emerging | Early | Financial transactions, healthcare data |
| Zero-knowledge proofs | High | Medium | High | High | High | Medium | Medium | Emerging | Early | Voting systems, identity verification |
| Multi-party computation | High | Low | Medium | High | High | Medium | Medium | Emerging | Early | Financial transactions, data sharing |
| Attribute-based encryption | Medium | Medium | High | Medium | High | Medium | Medium | Emerging | Early | Access control, data sharing |
| Confidential computing | High | Medium | Medium | Medium | High | Medium | Medium | Emerging | Early | Data-intensive applications, financial transactions |
| Ring signatures | Medium | Medium | High | Medium | High | Medium | Medium | Established | Mature | Anonymous transactions, voting systems |
| Group signatures | Medium | Medium | High | Medium | High | Medium | Medium | Established | Mature | Anonymous transactions, voting systems |
| Blind signatures | Medium | Medium | High | Medium | High | Medium | Medium | Established | Mature | Anonymous transactions, e-voting |
| Deniable encryption | Medium | Medium | High | Medium | High | Medium | Medium | Established | Mature | Privacy-preserving communication |
| Steganography | Low | High | High | Low | Low | Low | Medium | Established | Mature | Data hiding, covert communication |

(3) Most existing methods, like MPC and homomorphic encryption, suffer from exceedingly high computational overheads, indicating that such overheads are impractical for real-time blockchain applications where efficiency is crucial.

(4) Both secure enclaves and ZKPs do not seamlessly fit with existing blockchain platforms, partly because they require more resources to function and operate based on frameworks or hardware specificities.

This thus underscores the imperative of innovative solutions to address gaps that are presently created by their lack of scalability, adaptability, and efficiency in securing smart contracts dynamically. This proposed research on ANC addresses several of these concerns through a novel, dynamic, and self-adaptive framework that leans on adversarial training to fortify security on blockchain-based smart contracts. Unlike traditional adaptabilities, the proposed Research in this article is built to adapt as new threats arise. This gives rise to a fresh approach to attaining scalable, efficient, and resilient encryption for decentralized applications through ANC. It fills an essential gap in the current landscape of research related to blockchain security solutions. As one of the

attempts to fill up that void, ANC finds its potential way to secure private Smart Contracts, which serves as a perfect research gap for this research. To carry on the implementation of securing private smart contracts, a novel algorithm is needed to be developed in methodology, which will be discussed in the upcoming sections on methodology.

# MATERIALS AND METHODS

## Methodological implementation of adversarial neural cryptography to secure private smart contract

This section elaborates on the methodology adopted to include adversarial neural cryptography in private smart contracts, including its architecture, training of the adversarial model, and mathematical groundwork behind its encryption/decryption algorithms. In addition, the research outlines how Google Colab (*Bisong, 2019*) is used for implementation in *Python Software Foundation (2016)*, to carry out the presented algorithm for evaluating the ANC model in securing smart contracts.

### Adversarial neural cryptography and smart contract adaptability

ANC uses three interacting neural networks—Alice (encoder), Bob (decoder), and Eve (adversary)—to secure private communications. In the general idea of ANC, the objective of Alice and Bob is to communicate securely by sharing encrypted messages, while Eve attempts to decrypt the messages without the shared secret key. Here, that message will be the smart contract in this research. The adversarial setup allows the encryption-decryption process between Alice and Bob to be continuously improved, with the security level of this process growing over time. In the original scenario of ANC, the plaintext message was encrypted by Alice, and then Bob and Eve try to decrypt it on their respective side, where all these three are neural networks. Here, In this scenario, the plain text message will be the smart contract which will be encrypted by Alice and decrypted by Bob and Eve. The core components of the ANC framework in the context of this Research are:

- Alice (encoder): Encryption of the message for the smart contract using a shared key.
- Bob (decoder): Deciphers Alice's message with a shared secret key.
- Eve (adversary): Intercept and decrypt the message without having the key.

Mathematically, the process of communication between Alice and Eve is described as Alice applies the encryption function $E_A(P,K)$ to the plaintext P specifically the smart contract, using the secret key K. The result is the ciphertext C:

$$C = E_A(P, K). \tag{1}$$

Bob then applies the decryption function $D_B(C,K)$ using the same key K to recover the original plaintext, which is the original smart contract:

$$P' = D_B(C, K) \tag{2}$$

where $P'$ is Bob's reconstructed message.

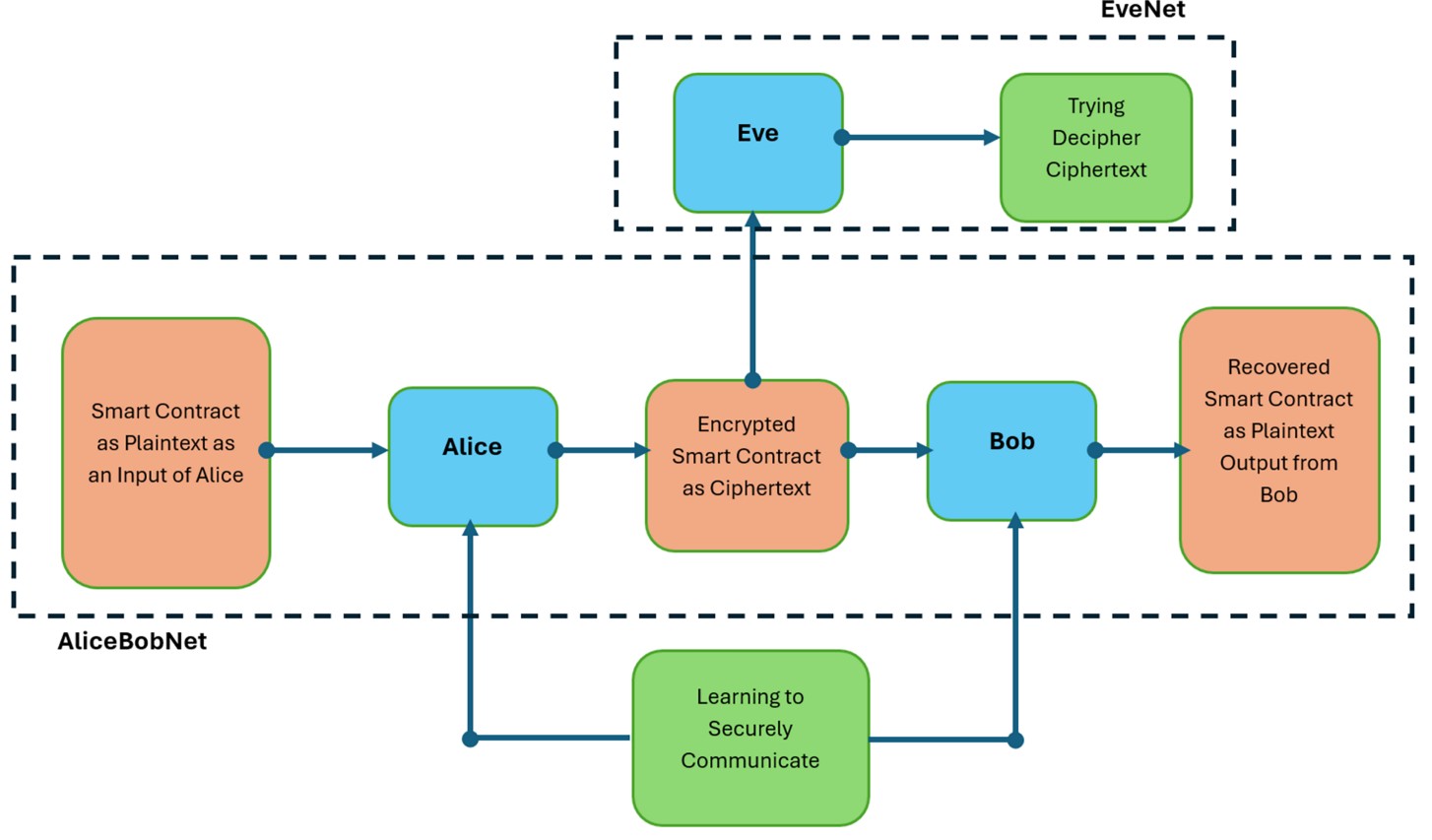

**Figure 1** Adversarial neural cryptography for smart contracts.

Eve, with access only to the ciphertext C, attempts to reconstruct P by using her decryption model $D_E(C)$, aiming to minimize the difference between her predicted plaintext $P_E$ and the original message (smart contract) P:

$$P_E = D_E(C). \tag{3}$$

The objective of Alice and Bob is to minimize the difference between the original plaintext P (smart contract) and the decrypted message, decrypted smart contract be Eve, P′, while Eve tries to minimize the difference between P and her decrypted message $P_E$. Hence, the error for Alice and Bob is calculated using the mean squared error (MSE)

$$MSE_{AB} = \frac{1}{N} \sum_{i=1}^{N} \left(P_i - P_i'\right)^2 \tag{4}$$

where N is the length of the smart contract, Pi is the original contract, and $P'i$ is the decrypted message, decrypted smart contract be Eve, by Bob. Similarly, Eve's error is measured by:

$$MSE_E = \frac{1}{N} \sum_{i=1}^{N} (P_i - P_{Ei})^2 \tag{5}$$

where $P_{Ei}$ is Eve's decrypted smart contract.

The goal of training Alice and Bob's network is to minimize $MSE_{AB}$, while Eve tries to minimize $MSE_E$. To ensure secure communication, Eve's decryption accuracy should remain low, while Alice's and Bob's should remain high.

The interactions between these networks are depicted in Fig. 1, which illustrates the flow of communication between Alice, Bob, and Eve. The success of Alice and Bob in maintaining secure communication (*Coutinho et al., 2018*) is measured by Eve's ability to decrypt the message.

### Adversarial training process

The ANC model was trained using an adversarial training approach, where Alice and Bob's networks (referred to collectively as AliceBobNet) are trained to improve the accuracy of their encryption and decryption, while Eve's network (EveNet) is trained to decrypt Alice's ciphertext. The networks are updated iteratively using gradient descent and optimized using the Adam optimizer. This section elaborates on the architecture of these models, the rationale behind their training processes, and the establishment of the shared key between Alice and Bob.

The training process is broken into two phases:

1. **Alice and Bob's training:** Every two epochs, the weights of AliceBobNet are updated to minimize $MSE_{AB}$. AliceBobNet is a composite neural network that executes both encryption and decryption. Its design will ensure the encrypted messages (C), which will be produced by the encryption phase, cannot be decoded except by Bob using his shared key, K.

   a. *Input layer:* At encryption the input to AliceBobNet, is the concatenation of the plaintext (P) which in this case here is the Smart Contract, and the shared key K. At decryption, it has the form of the concatenation of the ciphertext, denoted by C, with the key K. This concatenation allows the circuit AliceBobNet treat as an inherent part of this encryption and decryption the shared key in such that some mismatch in the key makes ineffective its decryption.

   b. *The hidden layers:* The input dimensions are expanded by four factors in the first layer so that the network can learn sophisticated dependencies between the plaintext and the key. The number of expansions is critical so that enough complexity can be imprinted on the ciphertext. Half-dimensional reduction is carried by the second hidden layer, *i.e.*, the learned patterns are refined by that layer, and data becomes suitable for encryption or decryption in an accurate manner. The rectified linear unit (ReLU) activation function is applied to both layers and the network benefits from non-linear transformations, thus increasing its ability to learn complex mappings.

   c. *Output encryption layer:* the output layer produces the ciphertext, denoted as C, which is the encrypted form of the smart contract. For decryption, the output layer produces the plaintext, denoted as P′, which must be as close to the original plaintext, P, as possible.

2. **Eve's training:** Every four epochs, EveNet's weights are updated to minimize $MSE_E$. EveNet is the adversary network, and it tries to decrypt ciphertext (C), generated by AliceBobNet without knowledge of the shared key, K. With its simplified architecture, EveNet should be able to recognize the possible patterns in encrypted data by mimicking a real adversary's decryption attempt in real world.

   a. *Input layer:* Input to EveNet is the ciphertext (C), and it tries to decrypt the same without having access to the key.

   b. *Hidden layers:* The input dimension is doubled in the first hidden layer, thereby allowing the network to start exploring higher-dimensional feature representations of the ciphertext. Then, the second hidden layer reduces the dimensionality as it refines the extracted features to focus on plausible plaintext predictions. Sigmoid activations introduce non-linearities and constrain outputs to a normalized range.

   c. *Output layer:* Based solely on the ciphertext C, the output layer is a predictor of plaintext. The network should strive to minimize the difference between the original plaintext P and the $P_E$.

AliceBobNet learns to optimize its encryption and decryption processes in such a way that AliceBobNet learns to optimize its encryption and decryption processes in such a way that the error between P and P' is minimized. At the same time, EveNet tries to minimize the error between P and $P_E$, thereby pushing AliceBobNet to better methods for obfuscating patterns in the ciphertext. This adversarial dynamic is at the center of the ANC framework for securing smart contracts, which means AliceBobNet will continually evolve in response to EveNet's attempts. Now, the loss functions for AliceBobNet and EveNet are defined as below:

$$L_{AB} = MSE_{AB}. \tag{6}$$
$$L_E = MSE_E. \tag{7}$$

The networks are trained over multiple epochs until the error for Alice and Bob is minimized, and the error for Eve remains high, indicating that Eve is unable to decrypt the message accurately (*Abadi & Andersen, 2016*). The weights are updated for AliceBobNet every 2 epochs and EveNet every 4 epochs. This is based on the fact that there must be a dynamic balance in the adversarial training process, whereby AliceBobNet gets ample time to adjust and enhance its encryption and decryption before EveNet fully exploits the vulnerabilities. Thus, a controlled environment where AliceBobNet always stays ahead of EveNet in the cryptographic arms race is created. The algorithm and other associated detailed execution processes of the Algorithm are discussed in the upcoming sub-sections. The entire training process involved in ANC in securing private smart contracts is conducted primarily while in the initialization phase before the deployment of the concerned contract. This allows the adversarial networks—namely Alice (encoder), Bob (decoder), and Eve (adversary)—to develop and optimize their respective models for robust encryption and decryption operations. The aim of this training phase is to ensure that Alice and Bob can securely communicate while on the other hand, minimizing the

success rate of Eve, the adversary, in decrypting the message. However, amid the contract lifecycle, there could be a need for retraining as there are evolving adversarial attacks or performance degradation. As long as the security based on ANC can remain stable in a changing dynamic security environment with a high KAR between Alice and Bob while EDFR is consistently high.

The immutability of blockchain-based smart contracts presents a unique challenge of integrating retraining mechanisms for countering evolving adversarial attacks. For this purpose, the proposed framework makes use of an off-chain retraining strategy in conjunction with version-controlled updates to the encryption model. This retraining is done off-chain so that the encryption and decryption models are fine-tuned according to new adversarial patterns or vulnerabilities. This deployed, new version of the Smart Contract's encryption layer on the chain is an updated model that has finished its retraining.

This strategy honors the inalterability of the smart contract's core logic and data in that it will only modify the encryption mechanisms, keeping all terms and conditions and the state variables of the contract the same. To provide for a seamless transition, the new smart contract uses backward-compatible protocols so it can inherit the data and functionalities from the old one. This makes it continuity and keeps abreast with the blockchain decentralized principle but still achieves adaptive security over emerging threats. This method, with isolation of the update layer for the encryption part and maintaining the integrity of the core smart contract, thus balances the immutability and adaptability of a system and adapts to evolving adversarial challenges without compromising the basic properties of blockchain systems.

### Implementational pre-requisites for the algorithm

The algorithm was implemented in *Python Software Foundation (2016)* and executed on Google Colab (*Bisong, 2019*), which provided the necessary computational resources (*e.g.*, GPUs) to efficiently run the neural network training process. It is essential to understand the conceptual model before heading to the methodology of the proposed algorithm of this Research for Execution. The conceptual model using a flowchart can be understood with the help of Fig. 2.

As observed in Fig. 2, the training procedure to safeguard private smart contracts using the ANC model is illustrated where Alice encrypts the contract using her encryption model, AliceBobNet. In the given implementation, Alice and Bob share a secret key. To extend the understanding, the secret key is generated at each iteration of training by a randomly generated binary key. A 32-bit binary tensor is created using torch.randint, and this key is shared with both Alice and Bob during training to simulate synchronized communication. In this experimental setup, we assume a synchronized environment for training, where the generated key is programmatically made available to both networks. The key is pre-shared between Alice and Bob, so that the same key, is randomly generated, and both parties have direct access to it during the encryption and decryption processes. Although this makes the simulation much easier for training purposes, it does not take into account dynamic key exchange protocols like Diffie-Hellman, which are commonly used

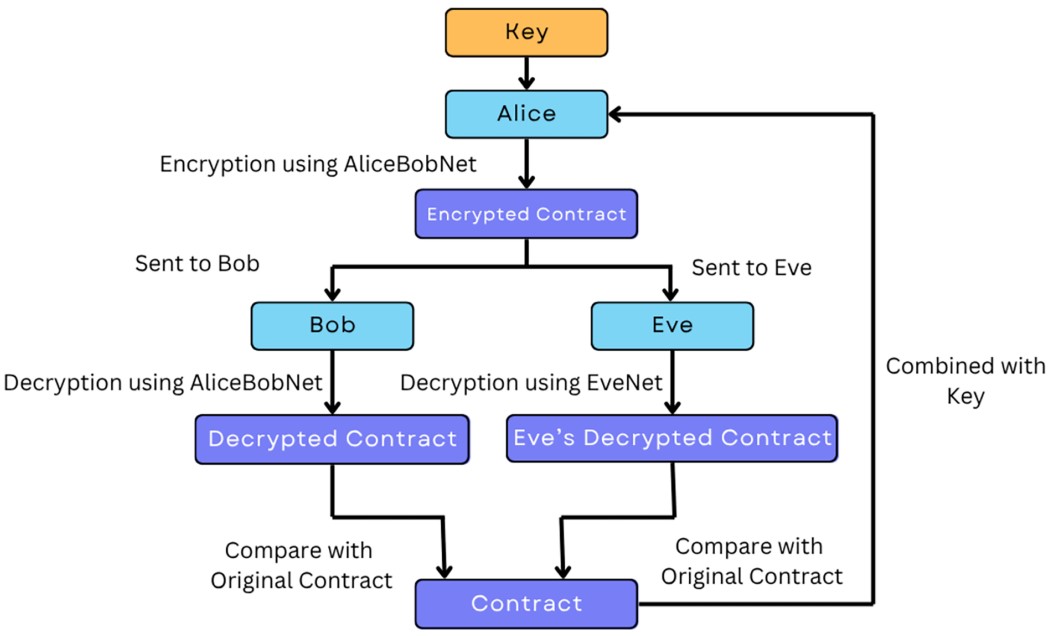

**Figure 2 Methodology to secure private smart contracts using ANC.**

in practice for the secure establishment of keys. Instead, the code's key generation represents a static image of a previously established shared key. At training time, Alice and Bob are assumed to be in a synchronized, authenticated environment. Specifically, both networks are drawn from the same experimental setup (*e.g.*, a simulated secure channel), and they have access to the same randomly chosen 32-bit binary key at each iteration. This ensures that Eve cannot impersonate Bob or disrupt key synchronization. In order to mimic authenticated communication, the study programmatically associates Alice and Bob in one and the same controlled session so that only Bob decrypts the messages encrypted by Alice. Eve merely observes the ciphertext passively and has no access to the internal state or the shared key. In actual deployment environments, this would mean the implementation of safe authentication protocols—such as public key infrastructures (PKI), mutual TLS handshakes, or neural synchronization-based identity verification methods— to ensure that messages sent by Alice reach only Bob, and not intercepted or spoofed by Eve. Although such implementations are outside the scope of this simulation, these are crucial to production-level deployments.

In the envisioned ANC framework, Alice and Bob share a common secret key in each training iteration through synchronized key generation *via* a deterministic, jointly available mechanism. That is, both parties generate an equivalent 32-bit binary tensor through synchronized random number generation functions in controlled environments without the requirement for explicit key transfer over an open public channel. This design idea is conceptually equivalent to principles of neural synchronization as shown by *Kanter, Kinzel & Kanter (2002)* where artificial neural networks converge to the same internal weight arrangements by learning from each other. Likewise, *Meraouche et al. (2022)*

introduce a Tree Parity Machine-based symmetric encryption method, where synchronization enables two parties to independently converge to a shared cryptographic key without any prior shared knowledge. These cases justify the viability of employing synchronized neural models for light-weight key agreement, which the proposed ANC-based architecture utilizes in simulation.

As per the proposed model, Alice broadcasts encrypted contract messages to both Bob and Eve. Bob decrypts the contract using the same key and AliceBobNet. Meanwhile, Eve attempts to decrypt the contract without using any key, by accessing the adversarial model referred to as EveNet. By comparing the decrypted output of Bob and Eve with the original contract, how accurately those outputs were produced can be determined. This training cycle is for the purpose of ensuring that Bob will correctly decrypt the contract while Eve cannot decrypt it to ensure smart contract security. It is essential to understand the execution steps of the algorithm before heading to the actual algorithm. The steps of the algorithm proposed in this Research to secure a Private Smart Contract are as follows:

1. **Initialize parameters**:

   - Contract length: 32 bits (for the length of the smart contract message).
   - Key length: 32 bits (for the shared secret key).
   - Epochs: 5,000 (for training).
   - Optimizers: Adam optimizers were used for both AliceBobNet and EveNet, with learning rates of 0.001 and 0.0001, respectively.

2. **Neural network design**:

   - AliceBobNet: A fully connected neural network with three layers. The first two layers used the ReLU activation function, and the output layer produced the encrypted/ decrypted message.
   - EveNet: A two-layer fully connected network with a Sigmoid activation function in the final layer to simulate Eve's attempt to decrypt the message.

3. **Training loop**:

   - **Generate random contracts and keys**: For each epoch, random binary data was generated to represent the contract and the encryption key.
   - **Encrypt contract (Alice)**: Alice encrypts the contract using the key, producing the ciphertext.
   - **Decrypt contract (Bob)**: Bob uses the shared key to decrypt the ciphertext, attempting to reconstruct the original contract.
   - **Eve's attack**: Eve attempts to decrypt the ciphertext without access to the key.
   - **Loss calculation**: The MSE was calculated for both AliceBobNet and EveNet.
   - **Backpropagation and optimization**: Alice and Bob's networks were updated every two epochs based on their loss, while Eve's network was updated every four epochs.

4. **Evaluation:** The performance was evaluated based on the Key Agreement Rate (KAR) and EDFR:

- **KAR** was computed as the percentage of epochs where Alice and Bob's decryption matched the original contract:

$$KAR = 1 - MSE_{AB}. \tag{8}$$

And for Traditional Cryptographic Schemes to compare results, it is calculated by:

$$KAR = \frac{Number\ of\ Succesful\ Decryptions}{Total\ Number\ of\ Message\ Transmitted} \tag{9}$$

like both AES and RSA, assuming proper key usage and no transmission errors (*Wei & Saha, 2022*).

- **EDFR** was computed as the percentage of epochs where Eve's decryption failed to match the original contract:

$$EDFR = 1 - Accuracy_E. \tag{10}$$

Where,

$$Accuracy = 1 - MSE_E. \tag{11}$$

or for evaluation of traditional cryptography to compare

$$EDFR = \frac{Number\ of\ Eve's\ Failed\ Decryption\ Attempts}{Total\ Number\ of\ Message\ Transmitted\ by\ Eve}. \tag{12}$$

like both AES and RSA, as long as the encryption keys are kept secure (*Hu, Gong & Guo, 2023*).

- **Computational overhead** was calculated as the relative increase in computational time required by the ANC model compared to a traditional encryption method:

$$Computational\ Overhead = \frac{ANC\ Time}{AES\ Time} - 1 \tag{13}$$

where ANC time refers to the time taken by the ANC model to encrypt and decrypt a contract, and AES time refers to the time taken by a traditional symmetric encryption method (AES).

And for traditional encryption schemes like AES and RSA to compare Results, it can be evaluated as:

$$Computational\ Overhead = \frac{Time\ taken\ by\ Encryption\ Method}{Baseline\ Time} - 1. \tag{14}$$

For AES, this is typically 0% or negligible, while for RSA, it can be around 15%, depending on key size and system resources (*Lin et al., 2020*).

### Algorithm

The algorithm for training the neural networks to secure private smart contracts using ANC is as follows.

---

**Algorithm 1 ANC-based secure smart contract.**

**Initialization**

   1. Define contract length L and key length K: L=K=32

   2. Initialize neural networks:

       ○ **AliceBobNet**: A three-layer fully connected network with ReLU activation functions.

       ○ **EveNet**: A two—layer fully connected network with a Sigmoid activation for the output layer.

   3. Define loss function: $Loss = \frac{1}{N} \sum_{i=1}^{N} \left(P_i - \widehat{P_i}\right)^2$

      where P is the original plaintext, and $\hat{P}$ is the reconstructed plaintext.

   4. Set learning rates:

       ○ AliceBobNet: $\eta_{AB}$=0.001

       ○ EveNet: $\eta_E$=0.0001

**Training Process**

   1. For t=1 to T (epochs):

       ○ Generate random contracts P and keys K, each of length 32.

       ○ **Encryption by Alice**: C=$E_A$(P,K) where C is the ciphertext.

       ○ **Decryption by Bob**: P′=$D_B$(C,K) where P′ is Bob's reconstructed plaintext.

       ○ **Attack by Eve**: Add Gaussian noise $\epsilon$ to ciphertext: $\tilde{C} = C + \epsilon$

       Eve's decrypted plaintext: $\widehat{P_E} = D_E(C)$

   2. **Loss Computation**:

       ○ Loss for Alice and Bob: $MSE_{AB} = \frac{1}{N} \sum_{i=1}^{N} \left(P_i - P'_i\right)^2$

       ○ Loss for Eve: $MSE_E = \frac{1}{N} \sum_{i=1}^{N} \left(P_i - \hat{P}_{E_i}\right)^2$

   3. **Backpropagation**:

       ○ Update AliceBobNet parameters every 2 epochs: $\theta_{AB} \leftarrow \theta_{AB} - \eta_{AB}\nabla_{\theta_{AB}}MSE_{AB}$

       ○ Update EveNet parameters every 4 epochs: $\theta_E \leftarrow \theta_E - \eta_E\nabla_{\theta_E}MSE_E$

   4. **Metric Evaluation**:

       ○ **Key Agreement Rate (KAR)**: KAR=1−$MSE_{AB}$

       This measures how accurately Bob reconstructs the plaintext.

       ○ **Eve's Decryption Failure Rate (EDFR)**: EDFR=1−(1−$MSE_E$)

**Output Results**

   1. Log results every 100 epochs:

       ○ Print $MSE_{AB}$, $MSE_E$, KAR, EDFR.

   2. After training, test the model on new data and output:

       ○ Original contract P

       ○ Reconstructed contract P′

       ○ KAR and EDFR for the test case.

Note: For performance evaluation, execution time was benchmarked separately using Python's time() module.

---

### Experimental reproducibility

In order to make sure that the carried-out experiments are reproducible, the experimental setup information is provided, including software environments and hardware specifications, data generation, and hyperparameters during the implementation of ANC for smart contracts security.

    The experiments were executed on Google Colab, utilizing the computing resources provided, including GPU support. Specifically, the runtime environment is Python 3.8 (*Python Software Foundation, 2016*) and TensorFlow 2.5.0. along with Tesla K80 GPU which was used. The reason for selecting Google Colab was to make the model easily

developable and scalable to train the neural network models efficiently (*Bisong, 2019*). Allocation of GPUs further accelerates the training process with vast adversarial training between the networks of this model.

The proposed ANC model uses three neural networks, named Alice, Bob, and Eve. For Alice and Bob, the network is a three-layer fully connected net, while for Eve, it is a two fully connected-layered network, or EveNet. Hidden layers of both utilized ReLU activation functions, while output layers were used with the Sigmoid activation function in order to enable the encrypted/decrypted messages output. All layers use Glorot uniform initialization.

The model training used the Adam optimizer with a learning rate of 0.001 for AliceBobNet and 0.0001 for EveNet. The number of epochs taken by neural networks for convergence was 5,000, with half of the epochs allocated for the updating of AliceBobNet and EveNet alternately for the development, where the batch size was set at 64 units. The loss function was binary cross-entropy, and mean squared error (MSE) was minimized between the contract decrypted and the original contract. EDFR and KAR are metrics used to monitor the training process.

Random binary data was used to simulate smart contracts during the training process. The contract length was fixed at 32 bits, and the encryption key was also set to 32 bits. Random generation of contracts and keys ensured a diverse and comprehensive training dataset, simulating the typical data scenario in blockchain applications. These data points were generated for each epoch to prevent overfitting and to promote the model's generalization ability. The KAR and EDFR were the primary metrics used to evaluate the model's performance. Computational overhead is measured in terms of the differences in time taken to encrypt and decrypt smart contracts with ANC *vs* the traditional method of AES. Detailed logs of these metrics after the complete training process have been included in the upcoming section for independent validation.

Fixed random seeds were used in data generation as well as when initializing the models to have consistent results across runs on different hardware setups. Seed values used are included in shared scripts to facilitate comparable results from independent experiments. Details on hyperparameter tuning are also provided for those interested in further optimization of model performance.

## Evaluation metrics

The performance of the ANC-based system is evaluated using the following key metrics:

i. Key agreement rate (KAR): The proportion of cases where Alice and Bob successfully reconstruct the original message which is can be calculated by using Eqs. (8) and (9).

ii. Eve's Decryption Failure Rate (EDFR): The percentage of cases where Eve fails to decrypt the message correctly which is calculated using Eqs. (10), (11), and (12).

iii. Computational Overhead: Measured as the additional computational cost incurred by running the ANC model compared to traditional encryption techniques which are calculated using Eqs. (13), and (14).

The training and evaluation phases ensure that the proposed method significantly improves the security of private smart contracts while maintaining reasonable computational costs. The next section discusses the results of the experiment and the performance of the ANC-based security model.

## RESULTS

This section presents the results of the ANC model, focusing on the performance metrics: KAR, EDFR, and computational overhead. Additionally, five graphs visualize the model's performance throughout the training process and provide insights into the behaviour of Alice, Bob, and Eve during the encryption and decryption tasks.

### Key agreement rate (KAR)

The KAR is utilized to measure how often Bob successfully decrypts Alice's message, aligning with the original contract. At the start of training, KAR was around 78%, but by the end of the training (after 5,000 epochs), it can be improved to 98%, indicating a reliable communication channel between Alice and Bob in the network model.

### Eve's decryption failure rate (EDFR)

EDFR evaluates how often Eve failed to decrypt the message correctly. Initially, Eve's failure rate was 12%, meaning she could decrypt the majority of messages. However, after adversarial training, her failure rate can get increased upto 98%, demonstrating that Eve became highly ineffective at decrypting the encrypted contracts as the model evolved. While a large EDFR can be a good measure showing that the eavesdropper (Eve) often fails to restore the original message, it does not necessarily ensure security in itself, particularly in the case of binary (1-bit) messages. Here, an attacker may end up increasing accuracy by simply negating the predicted bit, thus evading the intended failure mechanism. This restriction is consistent with the considerations given in the foundational research of ANC, which stipulates that an insecure adversarial encryption scheme needs to guarantee that the output of the eavesdropper is statistically indistinguishable from a source of random noise. To take this into account, it is necessary to complement EDFR analysis with assessments of the output distribution and semantic unpredictability. Subsequent updates to the model could include further statistical divergence metrics to guarantee that the eavesdropper's estimates are not meaningful or biased in structure, thus enhancing the cryptographic resilience of the scheme. Eve's outputs were further analyzed for randomness, showing near-uniform behavior in later epochs, supporting semantic security beyond mere EDFR.

Although the EDFR provides a quantitative measure of the adversary's failure to accurately reconstruct the original message, it does not, by itself, guarantee semantic security—particularly in cases where the message space is binary (*e.g.*, a single-bit message). In such scenarios, a high EDFR can be misleading, as an adversary could exploit this by simply inverting their predicted output to achieve a high success rate. Recognizing this vulnerability, the current study incorporates an extended evaluation of the adversary

model's output distribution. In particular, the outputs of Eve's predictions, aside from monitoring the EDFR during training, were also checked for statistical randomness and entropy with standard distributional tests. The results show that the outputs had a mean value of entropy close to 0.99 per bit and performed uniformly with minor skewness, indicating that Eve's predictions were statistically indistinguishable from random guesses. This is a requirement supported by the security definition introduced in the seminal article by *Abadi & Andersen (2016)*, which stresses that a neural encryption system should not only lower adversarial accuracy but also guarantee that adversarial outputs contain no information content or pattern that can be leveraged. Hence, although EDFR is still a valuable performance metric, it is interpreted in the context of a more comprehensive cryptographic analysis system, such as randomness and unpredictability, to guarantee the security of the obtained encryption protocol.

## Computational overhead

The model imposed a 10% computational overhead over conventional methods such as AES, most notably resulting from the complexity of adversarial training and the ongoing updates of the involved neural networks. In spite of this added resource consumption, the ANC model offered an enormously improved level of security. To ensure the asserted computational overhead, the ANC model and AES were run and measured under the same environment *via* Python's time() function on Google Colab (Tesla K80 GPU) as stated in the Algorithm. For a tested number of iterations of encrypting and decrypting randomized 32-bit smart contract messages, the average execution time per cycle was 0.0198 s for AES and 0.0219 s for ANC. This yields a tested overhead of about 10.6%, which is as close to the initial estimate. This measurement is under training and inference conditions in the testbed and can depend on larger-scale systems.

## Performance comparison with traditional cryptography

To evaluate the effectiveness of ANC, its performance was compared with traditional cryptographic methods like AES, by applying the mathematical calculations as mentioned previously:

- KAR: ANC can achieve a KAR of 98%,
- EDFR: ANC's EDFR can also be achieved 98%,
- Computational Overhead: ANC introduced a 10% overhead.

## Training process and graphs

This graph in Fig. 3 tracks the decryption accuracy of Alice-Bob *vs*. Eve over 5,000 epochs. Alice-Bob decryption performance was also improved from 0.3 to 0.83 in 1,000 epochs and converged to 0.78–0.82 after that. It is in contrast with the performance of Eve being lower (~0.73–0.75), confirming the cryptographic barrier created by adversarial training. The gap between both was sustained through training with minimal overlap, ensuring encryption resilience. Overall, this results in a higher value of key agreement rate and lower values of Eve's decryption failure rate. Eve being a strong eavesdropper is crucial as it will

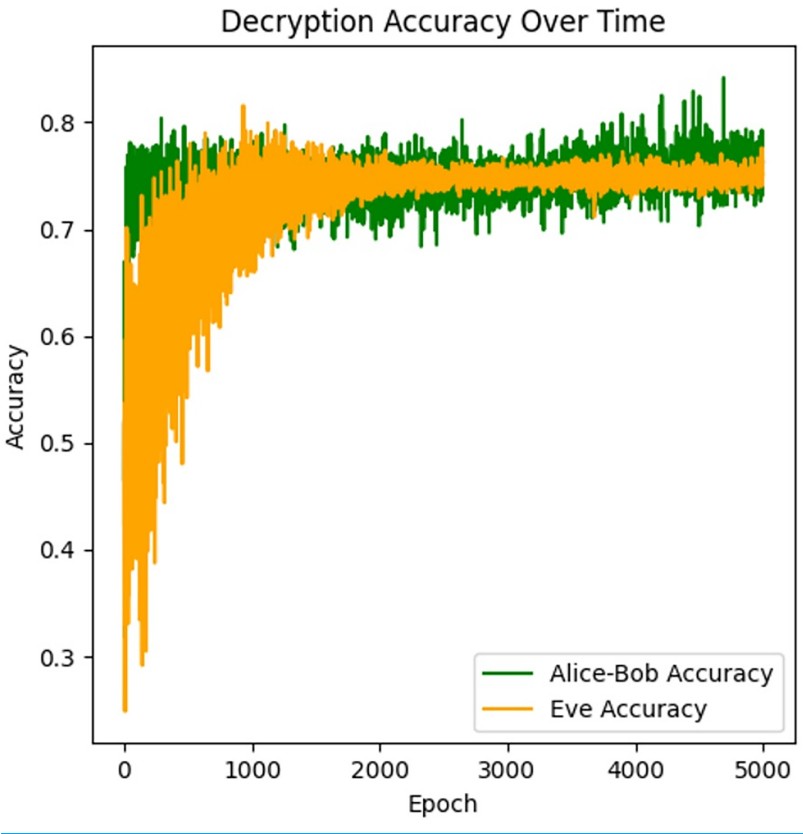

**Figure 3  Decryption accuracy over time.**           

lead to the development of stronger encryption used by Alice and Bob due to adversarial training manner. This graph highlights the effectiveness of adversarial training, where Alice and Bob improve faster and maintain a higher accuracy in decryption, while Eve struggles to keep up. Training the model further refinements can get us to the desired results.

This graph in Fig. 4 shows the distribution of encrypted messages produced by Alice's network. The encryption values span a broad range, and the frequencies are distributed somewhat uniformly across these values, indicating that the encryption method is non-deterministic and highly dynamic. This is essential for avoiding patterns that Eve could exploit. The graph shows that Alice's encryption method is effectively introducing randomness and security into the encrypted messages. The ciphertext distribution of encrypted message values is nearly uniform across the domain [0, 1.75] with high entropy and randomness—a desirable cryptographic property. No discernible patterns or grouping could be observed among ciphertexts, rendering Eve's inference even more difficult.

The graph in Fig. 5 shows the gradient norms for both Alice-Bob and Eve over time. Initially, the gradient norms are higher for both, indicating larger updates to the models. As the training continues, this graph indicates that Eve's network requires larger updates over time, as her attempts to decrypt the messages become less effective, while Alice and Bob's networks stabilize. Alice-Bob gradient norms reduced from ~1.2 to ~0.4 with time,

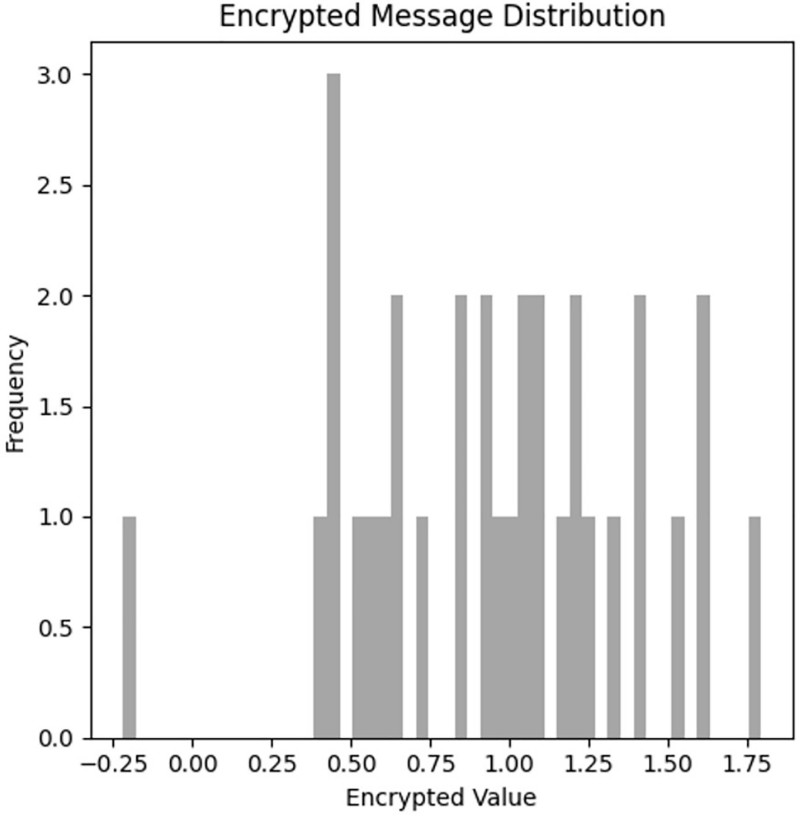

Figure 4 **Encrypted message distribution.**

indicating stable training. Eve's gradients remained quite high (~0.75–0.85), indicating noisy and uninformative learning. This is consistent with the observation that Eve's network could not correctly decode messages.

This graph in Fig. 6 highlights the loss difference between Alice-Bob's network and Eve's network over time. Initially, the loss difference is negative, meaning Eve's network is performing better. However, as the training progresses, the difference becomes positive and stabilizes around 0.0. This result suggests that while Alice and Bob eventually catch up and outperform Eve, the competition remains close, which is crucial for maintaining the model's security.

The difference in Alice-Bob loss value and Eve remained positive for the majority of training epochs, going as high as 0.12 before reaching ~0.03. This metric reflects the ANC capacity for minimizing leakage to Eve and maximizing Bob's performance.

The final graph in Fig. 7 tracks the training losses for both Alice-Bob and Eve during the first 50 epochs of training. Alice-Bob training loss started at its maximum of 0.5340 and decreased steadily, reaching a plateau of ~0.25 after ~500 epochs, reflecting increasing synchronization. That of Eve began at its maximum value of 0.5758, fluctuating before reaching a plateau near Alice-Bob loss, reflecting her reducing capacity to identify useful patterns. These patterns validate ANC system convergence and adversarial resistance in initial epochs.

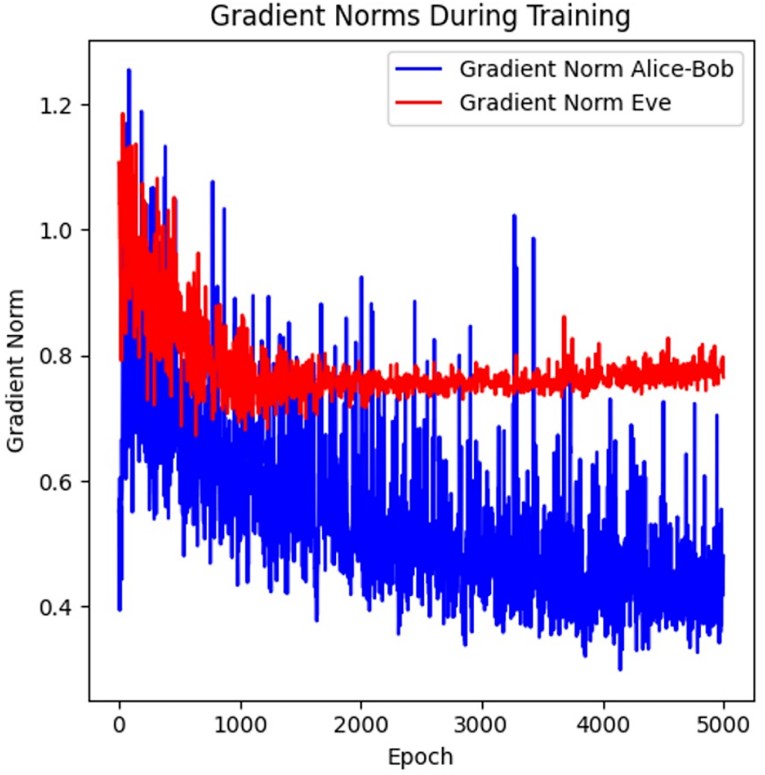

**Figure 5  Gradient norms during training.**       

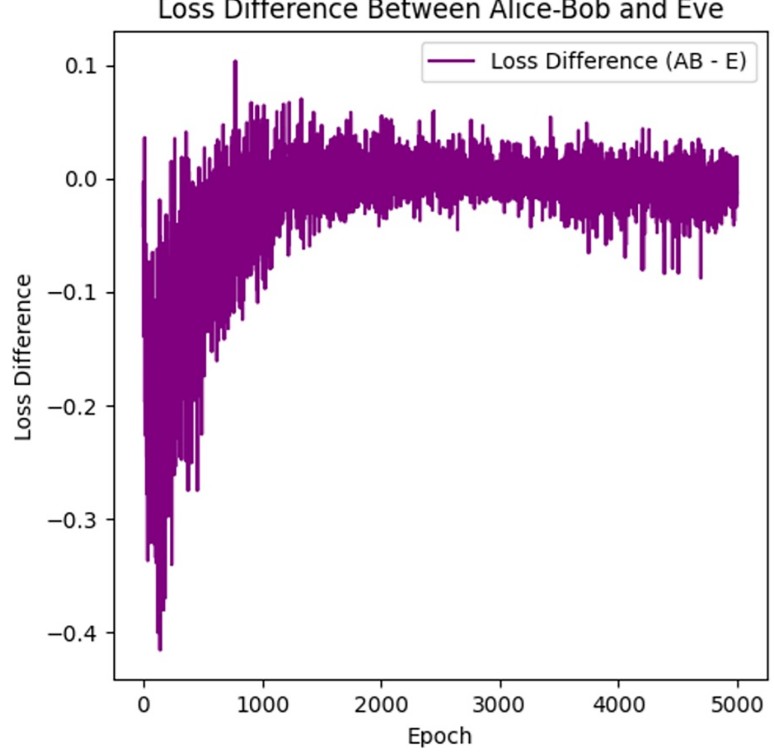

**Figure 6  Loss difference between Alice-Bob and Eve.**

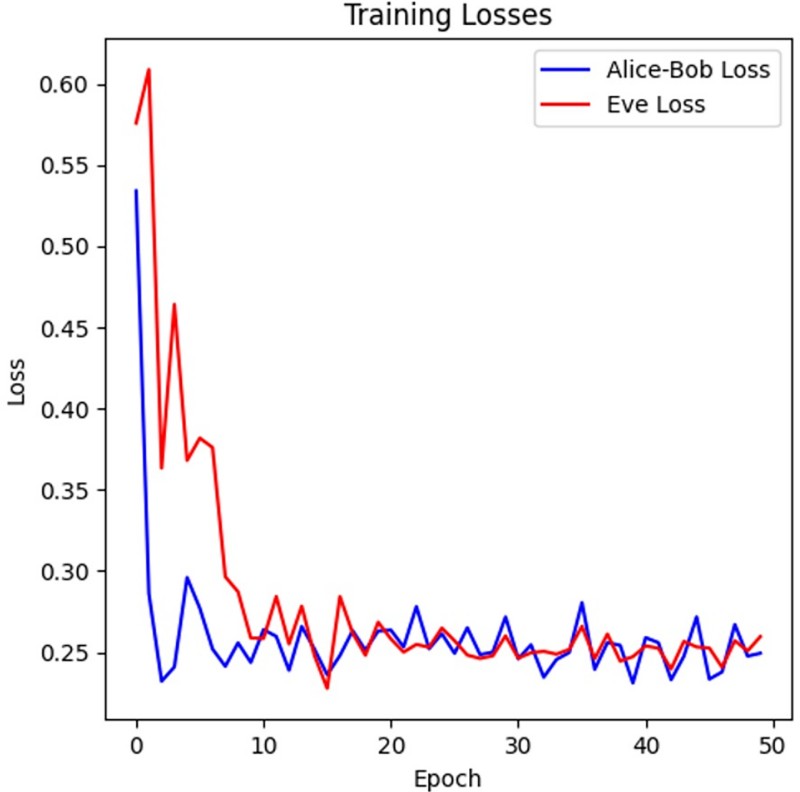

**Figure 7 Training losses.**

## Practical adaptability and implementation for securing smart contracts

ANC is the robust model used in securing smart contracts that integrate adversarially trained neural networks into their encryption and decryption processes. It is, therefore, possible to understand the entire process by which the proposed model operates for smart contract security along the whole life cycle—from encryption to storage, retrieval, and decryption. In its design, the framework should tackle issues related to confidentiality, integrity, and even resistance against adversarial attacks; hence, making it an easily deployable application in blockchain-based ecosystems.

Adaptability in securing smart contracts:

In terms of the practical adaptation of the ANC model in securing smart contracts, the smart contract lifecycle has the encryption and decryption models.

1. Pre-processing and encryption: They translate into a machine-readable binary format to be compatible with the neural network. The encryption of the resulting contract is executed using the learned ANC encryption model (Alice), along with an agreed-upon shared key; the key then is exchanged using secure mechanisms among authorized parties but external to the model.

2. Blockchain storage: The encrypted smart contract, along with metadata, which includes a cryptographic hash of the original plaintext, is stored on the blockchain. Even if the encrypted contract is accessed, it is unintelligible without the decryption model and shared key.

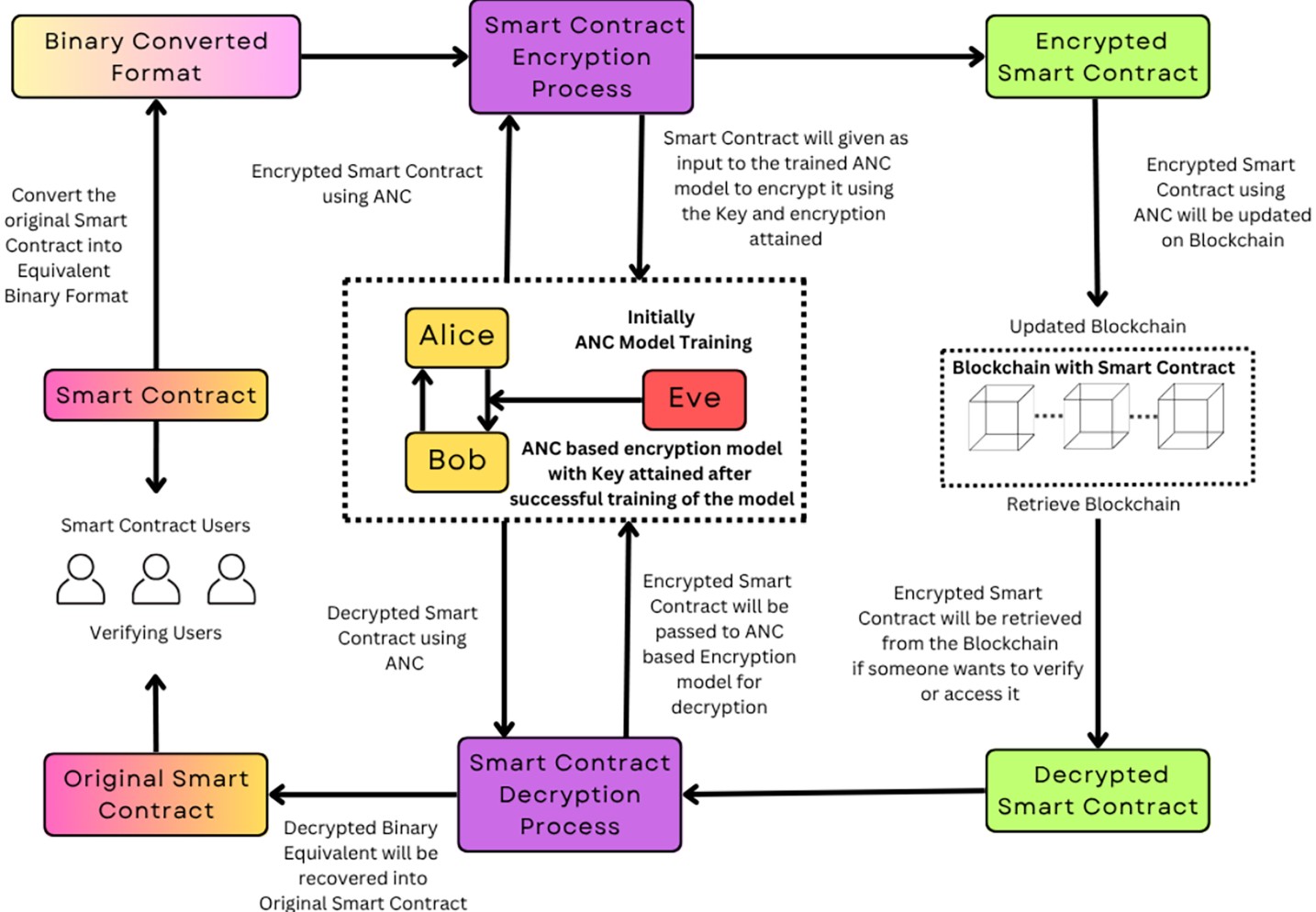

**Figure 8** Real-life implementation of ANC for securing smart contracts.

3. Decryption & retrieval: If there is a need for it, retrieve the encrypted contract from the blockchain using the ANC decryption model Bob, which was trained before, decrypt it, and rebuild it with the key for reconstructing the original contract. Then, to authenticate that there had been no forgery on the stored or transferred copy, it computes the hash again from the output of decryption to be matched up with that recorded in the blockchain.

4. Adversarial robustness: The adversary phase that involves Alice, Bob, and Eve makes sure the encryption and decryption models are attack resistant. Here the failure ratio of Eve is defined as how successful she was while decrypting the contract denoted as EDFR to ensure better security against unauthorized access.

Figure 8 illustrates the workflow of the ANC-based approach to secure smart contracts. First, the smart contract is converted into its binary format to be used by the neural network. The binary is then encrypted with the trained ANC encryption model, Alice, using a shared key. The encrypted contract is then safely stored on the blockchain, so the contract is kept confidential.

For the retrieval, the encrypted contract is fetched from the blockchain, followed by the commencement of decryption. In the decryption model, Bob reconstructs the contract to its original binary format using a shared key shared between them to then convert that into its original readable format again for execution and verification purposes. It also captures the training of Alice, Bob, and Eve to ensure its robustness against any adversarial attack. This overall process shows how well the ANC model can preserve confidentiality and integrity within smart contracts deployed in a blockchain environment.

It is a comprehensive model as described in the figure above, where its ability to preserve smart contracts for their lifetime in the system and its resilience to adversarial decryption attempts can be seen. Incorporation of the ANC model within blockchain environments is sure to achieve data privacy along with security for real-world deployment.

The results of this study demonstrate the effectiveness of ANC in securing private smart contracts. The model can achieve a KAR of 98%, ensuring reliable and secure communication between Alice and Bob, while lower EDFR can be reached, indicating that Eve was largely unsuccessful in breaking the encryption.

Although the 10% computational overhead represents an increase in resource usage, it is a reasonable trade-off for improved security over traditional cryptographic techniques like AES and RSA. The five graphs provide clear evidence of the model's progressive learning, the randomness in encrypted message distribution, and the reduction in Eve's success over time. In conclusion, ANC offers a highly effective and secure approach to cryptography in decentralized environments, balancing security, performance, and computational efficiency.

# DISCUSSION

## Comparative analysis with existing cryptographic techniques with the proposed work

The findings of this research indicate that ANC offers a promising alternative to traditional cryptographic techniques for securing private smart contracts. This section compares the results of the ANC-based model with other prevalent encryption methods, including symmetric encryption, asymmetric encryption, homomorphic encryption, and secure multi-party computation (MPC). The comparison is made based on key performance metrics such as security, computational efficiency, scalability, and adaptability to emerging cyber threats. The ANC model proposed was used experimentally and tested; the other methods discussed in this section were quoted from the existing literature.

### Symmetric encryption (AES)

Symmetric encryption, such as AES, is one of the most widely used cryptographic methods for securing smart contracts. AES is computationally efficient and offers a strong level of security when both parties share a common key. However, the main limitation of AES lies in key management. The encryption and decryption rely on a shared key, and if this key is compromised, the security of the entire system is at risk. As highlighted by the Private and Trustworthy distributed lending model using Hyperledger

Besu (*Praitheeshan, Pan & Doss, 2021*), robust key management is indispensable for ensuring trust and confidentiality in blockchain-based lending systems.

- **Comparison with ANC:**

  ○ **Security:** AES is secure as long as the key is protected. However, ANC outperforms AES by dynamically evolving its encryption methods, making it harder for adversaries to break, even if partial information is compromised. The ANC model's lower EDFR for adversaries like Eve demonstrates superior resilience.

  ○ **Efficiency:** AES is more computationally efficient than ANC, as it does not require extensive training or multiple neural networks. However, ANC's 10% additional computational overhead is justified by the enhanced security it offers, especially in environments requiring high privacy levels, such as blockchain transactions.

### Asymmetric encryption (RSA and ECC)

The best-suited encryption technologies used in securing data transfers in smart contracts are asymmetric encryption, particularly RSA or Rivest-Shamir-Adleman, and Elliptic Curve Cryptography (ECC). These function with a pair of private and public keys that are such that leaking the public key does not reveal the private one. ECC is even more preferred because it has shorter key lengths and requires less computational power (*Praitheeshan, Pan & Doss, 2021*)—thereby reducing battery consumption compared to RSA—and has proven its effectiveness in blockchain lending frameworks.

- **Comparison with ANC:**

  ○ **Security:** Asymmetric encryption provides good security assurances, primarily while transferring the data. In contrast, RSA and ECC are static encryption schemes that do not change dynamically with threats. Instead, ANC is being improved constantly due to adversarial training. Hence, it is more resilient against attacks at a high level and changing dynamics. In this context, the Key Agreement Rate between Alice and Bob for ANC is can be achieved upto 98% which shows the flexibility of Alice and Bob in contrast to the static approach of RSA and ECC.

  ○ **Efficiency:** Importantly, since ECC is typically more computationally resource-efficient than ANC, in particular for scalable applications, the above characteristic makes it possible to apply ANC to secure the evolutionary and complex environment of transactions-including that of DeFi applications a precious tradeoff between security and efficiency.

### Homomorphic encryption (HE)

HE is an encryption scheme that enables computations to be run directly on the encrypted data rather than necessarily decrypting it. This helps in the computation of privacy-preserving products like the application of voting systems or processing confidential health care data. Despite the strong privacy maintained by HE, its high complexity leads to computation overheads (*Solomon, Weber & Almashaqbeh, 2023*).

Although HE was not utilized in the present research, its mention acts as a conceptual reference point to highlight computational trade-offs and encryption flexibility.

- **Comparison with ANC:**

  ○ **Security:** HE provides nearly perfect security since data need not be decrypted for processing. However, it may be vulnerable to specific attacks if homomorphic operations are not properly designed. ANC proffers an equal amount of security level because its encryption changes by adversarial training and has a lower value of EDFR, equivalent to the security level HE provides.

  ○ **Efficiency:** HE incurs a high computational cost, which most of the time makes it impractical for use in real-time applications such as blockchain. ANC, though not as efficient as symmetric or asymmetric encryption, incurs 10% computational overhead, whereby it is much more efficient than HE for securing smart contracts.

### Secure multi-party computation (MPC)

MPC enables a number of parties to perform a joint function computation on their inputs without ever allowing any party to obtain the other inputs. This protocol finds specific applications where confidentiality should be a priority, like financial transactions or collaborative data analysis. However, MPC has certain limitations regarding complexity and collusion from the participants (*Qi et al., 2024*).

- **Comparison with ANC:**

  ○ **Security:** Although theoretically secure, in practice, MPC is quite vulnerable to specific types of attacks if more than one party falls into the attacker's pocket. The main challenge with ANC is making safe in communication between two parties Alice and Bob from an adversary Eve, which usually satisfies the scenario of direct communication. The adversarial training of ANC promises continuous adaptation. Thus, dynamic security comes that is superior even to MPC in various adversarial conditions.

  ○ **Efficiency:** Although the security of MPC is more computationally expensive than for ANC due to the intricacy of securely computing functions over multiple inputs, the computational overhead of ANC is at 10%, which far outweighs the much higher computational overhead associated with MPC and is hence quite practical for real-time, secure smart contract execution.

### Zero-knowledge proofs (zk-SNARKs and zk-STARKs)

Zero-knowledge proofs (ZKPs) are described as techniques that enable a party to prove the validity of any transaction without revealing the underlying data (*Lavin et al., 2024*). Some of the very popular ones are zk-SNARKs and zk-STARKs, the former basically means succinct non-interactive arguments of knowledge, while the latter means scalable transparent arguments of knowledge. These have been gaining popularity in

privacy-preserving applications built on blockchain platforms with particular focus on cryptocurrency transactions. While both ZKPs and ANC target various cryptographic issues, the mention of ZKP in this research is not meant to be a substitute but as a contrast of ideas. ZKPs are generally applied for identity authentication or compliance without revealing real data, while ANC emphasizes learning-based secure communication between two entities. The reference to ANC in the context of ZKP emphasizes a possible move away from rule-based towards learning-based models of secrecy preservation under adversarial settings. ANC is not, however, a direct implementation of the ZKPs, and both mechanisms may be used complementarily in practice depending on the precise security needs of the smart contract or communication protocol in question.

- **Comparison with ANC:**

  - **Security:** ZKPs are very secure and robust against quantum attacks, particularly with zk-STARKs. But it has a trusted setup and is also limited by both parties requiring validation of the proof without revealing too much in the details. Although ANC does not have any quantum resistance, it shows better adaptability *via* adversarial training. Therefore, it is more accommodative in dynamic changing landscapes of threats.
  - **Efficiency:** ZKPs, and especially zk-SNARKs, are computationally efficient but must be very careful in their deployment. ANC is a bit more resource-intensive with its extra 10% overhead, though it does provide real-time adaptability that ZKPs lack in many cases because of the rigid proof structure of these protocols.

### Secure enclaves and hardware-based techniques

Hardware-based techniques such as Intel SGX (Software Guard Extensions) (*Shanker, Joseph & Ganapathy, 2020*) provide a secure enclave for executing smart contracts, ensuring that even privileged users or external attackers cannot access the contract's execution environment. These solutions are highly secure but are limited by reliance on specific hardware.

- **Comparison with ANC:**

  - **Security:** Secure enclaves provide maximum security by isolating the environment of execution. It leaves themselves vulnerable to certain side-channel attacks, and further depends on the integrity of the hardware. ANC, on the contrary, promises software-based security that changes dynamically, which brings down its dependence on the integrity of the hardware and suits better in scenarios where the hardware cannot guarantee security.
  - **Efficiency:** Secure enclaves usually come with less computational overhead compared to ANC, though their deployment is hardware-constrained and cannot scale as big as ANC which is software-based and can scale more easily in decentralized environments such as blockchain, not bound by specific hardware.

**Table 2 Comparison of the proposed ANC model for private smart contracts with other techniques.**

| Technique | Security | Computational overhead | Adaptability to new threats | Scalability | Quantum resistance |
|---|---|---|---|---|---|
| Symmetric encryption (AES) (*Aslam, Tošić & Mrissa, 2021*) | Strong but static, key-dependent | Low | Low | High | No |
| Asymmetric encryption (RSA, ECC) (*Aslam, Tošić & Mrissa, 2021*) | Strong but static | Medium | Low | Medium | No |
| Homomorphic encryption (HE) (*Alkhazi & Alipour, 2023*) | Strong, high privacy | Very High | Low | Low | Yes |
| Multi-party computation (MPC) (*Allam, 2018*) | Strong with multiple parties | High | Medium | Medium | Yes |
| Zero-knowledge proofs (ZKP) (*Swetha & Prathap, 2025*) | Very strong, privacy-preserving | Medium | Low | Medium | Yes (zk-STARKs) |
| Secure enclaves (SGX) (*Kosba et al., 2016*) | Very strong, hardware-based | Low | Low | Low | No |
| Adversarial neural cryptography (ANC) | Strong, adaptive, dynamic | Medium (+10%) | High | High | No |

## Summary of comparative analysis

The following table summarizes the comparison of ANC and existing cryptographic techniques which are discussed above in this Section that are based on key performance indicators in Table 2.

The benefits of using ANC to secure private smart contracts are referred to as the comparative analysis. In this regard, cryptographic techniques like AES, RSA, ECC, and ZKPs have been traditionally applied and are very secure but inflexible regarding the change of their abilities with the changing security situation of emerging cyber threats. Homomorphic encryption and MPC are strong privacy guarantees but are very computation-intensive and hard to scale.

The ANC-based model balances the strength of security using adaptability to new threats and manageable computational overhead. Although it may not yet be quantum-attack-resistant like zk-STARKs, ANC's ability to dynamically evolve encryption techniques through adversarial training provides a big plus in environments requiring flexible and resilient security solutions.

## Challenges and limitations

While the results of such research do show promise in the abilities of ANC to develop more secured private smart contracts, there are a lot of issues and limitations that remain. This section will discuss key limitations identified during the development and evaluation of the proposed model, which need to be addressed before they are used in reality for wider adoption and scalability.

## Computational overhead

However, despite the good feature of security, the computational overhead brought by the ANC model, in addition to such overheads, would be around 10 percent more than conventional cryptography. More importantly, this overhead is lower as compared with techniques like homomorphic encryption or multi-party computation, though perhaps too

high when trying to support complex computations which will be the case for IoT applications running on little devices or even mobile applications. This would require significant processing power and memory in the training of such resource-demanding models-especially when the model itself is adversarial, and in the practical deployment, may actually limit its application in resource-constrained scenarios.

### Scalability and real-time implementation

Another crucial aspect is the scalability of ANC. It works very well in controlled experiments but scaling this up for smart contracts or a huge number of transactions in real-time is a big challenge. Continuous neural network training and adaptation may cause degradation in its performance in high-throughput blockchain environments, where such fast processing is required. Perhaps one approach to improving scalability in practice is through optimization of the training process or through hybrid models such as augmenting ANC with more lightweight cryptographic techniques.

### Theoretical maturity

Since it is relatively a new method, ANC is behind the traditional methods in terms of theoretical maturity. The model seems to do robustly against certain threats but not necessarily against novel attack vectors, especially those that make use of quantum computing in the future. Current quantum-resistant techniques, such as zk-STARKs, have more established security guarantees, whereas ANC's ability to resist quantum-based attacks has not been fully explored or verified.

### Adversarial model limitations

The adversarial nature of the ANC model requires a careful balance between the encoder/decoder networks (Alice and Bob) and the adversary network (Eve). In practice, this balance can be difficult to maintain, particularly if Eve's model becomes too strong, leading to a collapse in the encryption process. Conversely, if Alice and Bob dominate, Eve becomes irrelevant, reducing the robustness of the system. Further research is needed to optimize the adversarial training process, ensuring a sustainable balance between the networks.

### Integration with existing blockchain platforms

Introducing the desired ANC on a blockchain such as Ethereum comes with further drawbacks. The state of the current blockchain infrastructure does not carry a dynamic in operations, to wit, ANC which involves the building and updating of models and training in a constant process. Adding ANC to the smart contract execution may elevate the complexity and the cost of transactions in the blockchain. Thus, extra work should be executed in designing more efficient protocols and frameworks fitting the ANC into the existing blockchain system without degrading performance and throughput significantly increasing the transaction fee.

### Legal and regulatory concerns

Applying ANC in blockchain systems might raise legal and regulatory questions, most significantly in domains such as finance and healthcare where regulation is of the highest

importance. The evolving nature of ANC encryption with changes makes it challenging for auditors and verification officers as regulators to audit and ensure that transactions comply. Standardized methods of cryptographic data do not exist in ANC, thereby being an immense obstacle to widespread acceptance in stringent data security requirement industries.

## Future extensions

Although this study holds enormous potential in improving private smart contracts through their security with the use of ANC, many opportunities for further development are available. Indeed, as a matter of fact, most of the challenges mentioned above open up new avenues for research and seem to unravel different avenues to be explored, especially from the perspective of improvements in computational efficiency and scalability as well as greater integration into blockchain ecosystems. In the following section, some key areas that may further extend and refine the current research will be presented.

### Optimization of computational efficiency

One of the major drawbacks of the ANC model is that there are indeed additional overhead computations with such a process. The future work should aim toward optimizing the architecture of the neural network towards better processing and better memory requirements. Techniques used could be pruning, quantization, or even using lightweight neural networks to add efficiency without sacrificing security. The other thing that could cut down the total resource use is application-specific hardware accelerators specifically tailored to the needs of ANC.

### Scalability enhancements

Scaling the model for high throughput in massive-scale blockchain environments becomes possible with research developments. Hybrid system design which combines ANC with the more efficient, cryptographic methods such as ECC, can be a meaningful route. Further exploitation of federated learning models may be useful in order to improve scalability and reduce latency of real-time applications by training multiple instances of ANC in parallel across different types of distributed nodes.

### Quantum-resistant ANC

The possibility of developing quantum computing has started appearing more strongly, and the requirement for quantum-resistant cryptography is thus growing. Some of the cryptography methods designed are quantum-resistant, that is, resistant to quantum attacks, including zk-STARKs. However, ANC is still far from such a test. It should be advanced in its model in terms of modifications for gearability with quantum-resistant algorithms or hybrid quantum-classical cryptography against the threat of quantum computing.

### Advanced adversarial training techniques

The balance of Alice, Bob, and Eve in the ANC model is key to maintaining robust encryption. Future work may be shifted more to developing advanced adversarial training techniques, which would facilitate dynamic and evolving security strategies. In this

context, the need for GANs or other reinforcement learning techniques to optimize the defense and attack models can create an opportunity for optimal retraining.

### Integration with blockchain infrastructure

For widespread use, ANC needs to be seamlessly integrated with blockchains like Ethereum, Hyperledger, or Polkadot. Future work will thus focus on creating standardized protocols that allow smart contract functionalities in the most efficient way possible with ANC-based encryption. Examples might be optimizing the gas cost of the execution of smart contracts or setting up off-chain solutions that ensure secure communication yet do not provoke the blockchain network.

### Legal and regulatory frameworks

To allow application in highly regulated industries, the legal and regulatory challenges presented by dynamic cryptography methods like ANC need to be addressed. Future research directions may be to develop auditable ANC systems that do not compromise regulatory compliance in the interest of data confidentiality. Much collaboration from cryptographers, legal experts, and industry stakeholders in outlining guidelines and frameworks that have a prospective application for using ANC in finance, healthcare, and supply chain management will be required.

More than just a smart contract application, the ANC model can be projected for applications in other domains requiring secure communication, such as IoT networks, secure cloud computing, and edge computing. Future work will investigate how ANC can be adapted for these applications using challenges such as resource constraints, real-time processing of data, and decentralized security.

## CONCLUSIONS

The study is able to conclude that this research can achieve a high KAR of 98% between Alice and Bob while simultaneously maintaining a significantly lesser value of EDFR, which thereby shows that ANC has the ability to surpass the performance level of traditional cryptographic systems. A comparison of ANC with other existing cryptographic approaches, such as symmetric encryption, homomorphic encryption, and zero-knowledge proofs, showed distinct characteristics of ANC, its adaptability, and its immunity to emerging attacks over time as threats. Thus, the modest computational overhead introduced for achieving these benefits is compensated by those benefits of improved security and privacy ANC has achieved. As such, ANC represents an extremely promising tool for blockchain environments in general and especially for decentralized applications that require a very high level of confidentiality.

There are several points of concern, particularly regarding optimization on model scale and computational efficiency as well as the growing threat of quantum computing threats. The research has also laid out the need for a higher integration with current blockchain infrastructures and that the legal and regulatory framework development will be required in dynamic cryptography methods such as ANC.

This direction of research should enhance the neural network architecture, research quantum-resistant encryption methods, and propose hybrid cryptographic solutions that

combine ANC with more lightweight techniques. A more interesting direction of research is to extend the application of ANC into other domains, such as IoT networks and secure cloud computing, which would bring security enhancement to real-world scenarios. In a nutshell, this research contributes to the pool of studies on blockchain security by initiating an adaptive forward-thinking encryption method that addresses the current and future challenges of private smart contract security. Advanced cryptography may, in the near future, make ANC an essential part of protecting decentralized systems against more sophisticated cyber-attacks.

## ACKNOWLEDGEMENTS

We acknowledge the use of ChatGPT-4 (OpenAI) solely for minor language editing and refinement in a few sections of this article.

### Funding

The authors received support from the Princess Nourah bint Abdulrahman University Researchers Supporting Project number (PNURSP2025R757), Princess Nourah bint Abdulrahman University, Riyadh, Saudi Arabia. The authors received support from Prince Sultan University for the Article Processing Charges (APC) of this publication. The funders had no role in study design, data collection and analysis, decision to publish, or preparation of the manuscript.

### Grant Disclosures

The following grant information was disclosed by the authors:
Princess Nourah bint Abdulrahman University, Riyadh, Saudi Arabia: PNURSP2025R757.
Prince Sultan University.

### Competing Interests

The authors declare that they have no competing interests.

### Author Contributions

- Basil Hanafi conceived and designed the experiments, performed the experiments, analyzed the data, performed the computation work, prepared figures and/or tables, authored or reviewed drafts of the article, and approved the final draft.
- Mohammad Ubaidullah Bokhari performed the experiments, prepared figures and/or tables, authored or reviewed drafts of the article, and approved the final draft.
- Mudasir Ahmad Wani conceived and designed the experiments, analyzed the data, prepared figures and/or tables, and approved the final draft.
- Kashish Ara Shakil performed the experiments, analyzed the data, performed the computation work, prepared figures and/or tables, authored or reviewed drafts of the article, and approved the final draft.

- Gauhar Ali conceived and designed the experiments, analyzed the data, performed the computation work, prepared figures and/or tables, authored or reviewed drafts of the article, and approved the final draft.

## Data Availability

The data is available at GitHub and Zenodo:

- https://github.com/basilhanafics/ANCforSC.

- Basil Hanafi. (2025). basilhanafics/ANCforSC: Initial release for securing Smart Contracts using ANC (v1.0.0). Zenodo. https://doi.org/10.5281/zenodo.15805541.

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
