# Peer review of "Dynamic adversarial neural cryptography for ensuring privacy in smart contracts"

_PeerJ Computer Science, doi:10.7717/peerj-cs.3286_

## Round 0.1 · original submission · Major Revisions

· Academic Editor

Major Revisions

Reviewer 1 ·

Basic reporting

The authors have claimed that the model introduced a 10% computational overhead compared to traditional methods like AES, but no experimental result is provided to strengthen this claim. Most of the claims in the result sections made by the authors are just generic.

Experimental design

Experiments should be conducted to justify the claims made by the authors on how the computational overhead of the proposed model compares with other existing techniques. Also, authors have discussed Homomorphic Encryption and compared it with the proposed model. Have the authors implemented the HEs? If yes, I would like to see the pseudo code

Validity of the findings

-

Cite this review as

Reviewer 2 ·

Basic reporting

The paper under review tries to propose an alternative way to secure smart contracts using adversarial neural cryptography. The problem is interesting and worth pursuing. However, I would like to point out certain basic issues which are not expected from a research paper at this level.

a. It is not good to end a sentence with a citation.
b. “Although several cryptographic methods have already been adapted to secure these contracts, such as symmetric encryption and asymmetric encryption, the sophistication of these cyber threats is likely to demand even more advanced solutions for security in the future [4].” This sentence is an overshoot. Although, ANC is thought of possibly a quantum resistant primitive as there is no fixed, well-defined algorithm for encryption or decryption. However, lattice-based primitives which are well studied provide sufficient provable security guarantee against quantum adversaries. ANC is yet to face a rigorous testing against such quantum adversaries.
c. In several places capital letters are used in the middle of the sentences.

Other technical comments are as follows.

1. (Pg 17) “To extend the understanding, the secret key is generated at each iteration of training by a randomly generated binary key. A 32-bit binary tensor with the help of torch.randint is used to generate a key.” How do the nodes end up having the “same” secret key?
2. How does Alice know that it is training with Bob? And not with Eve? That is, how can the authors guarantee authenticated communication? Do they assume an authenticated channel between them which may not be secure?
3. The EDFR as defined in Pg. 19, (Line 557) requires to be very high, according to the authors, for the encryption rule to be secure. However, if we just consider one bit message (i.e., either 0 or 1) and EDFR is very high then here is an attack to the encryption rule : whatever the output of EVE is we will just flip the bit and we will be correctly decrypting the ciphertext with very very high probability. – I do not understand how the authors can bypass this attack. Generally, the EVE’s output should be close to uniformly random for an encryption scheme to be secure. Please look at Abadi-Anderson's fundamental work [5] for defining security.
4. It is not clear to me how the authors propose the usage of ANC in lieu of zero knowledge proof systems. The goals of the systems are different.
5. On several occasions, the authors claimed that Alice and Bob in the ANC model will have updated common secret keys in order to make the encryption-decryption training possible. I do not understand the process completely. I would also request the authors to look at the following paper : “Kanter, I., Kinzel, W., Kanter, E.: Secure exchange of information by synchronization of neural networks. EPL (Europhysics Letters) 57 (2002)” and also a follow up work by “Tree Parity Machine-Based Symmetric Encryption: A Hybrid Approach by Meraouche et al.”

Experimental design

The EDFR as defined in Pg. 19, (Line 557) requires to be very high, according to the authors, for the encryption rule to be secure. However, if we just consider one bit message (i.e., either 0 or 1) and EDFR is very high then here is an attack to the encryption rule : whatever the output of EVE is we will just flip the bit and we will be correctly decrypting the ciphertext with very very high probability. – I do not understand how the authors can bypass this attack. Generally, the EVE’s output should be close to uniformly random for an encryption scheme to be secure. Please look at Abadi-Anderson's fundamental work [5] for defining security. Consequently, the experiments based on this dubious/wrong pretext would result in severe errors.

Validity of the findings

I cannot validate the findings as there are too less detailing.

Cite this review as

---

## Round 0.2 · Major Revisions

· Academic Editor

Major Revisions

**Language Note:** When preparing your next revision, please ensure that your manuscript is reviewed either by a colleague who is proficient in English and familiar with the subject matter, or by a professional editing service. PeerJ offers language editing services; if you are interested, you may contact us at [email protected] for pricing details. Kindly include your manuscript number and title in your inquiry. – PeerJ Staff

Reviewer 2 ·

Basic reporting

The authors have updated their draft in a major way to answer the queries of the reviewers. The answers are mostly convincing. However, the authors are requested to do a major revision of the paper where --

1. The authors should make a separate section on smart contracts and their usage.
2. More literature review on smart contracts and
3. Make a separate contribution section on Page 7 and clearly point out what are the main challenges that the authors have overcome.

Experimental design

-

Validity of the findings

-

Cite this review as

---

## Round 0.3 · accepted · Accept

· Academic Editor

Accept

Dear Author,

Your paper has been accepted for publication in PeerJ Computer Science. Thank you for your fine contribution.

Reviewer 2 ·

Basic reporting

I believe the questions and suggestions that were posed by the reviewer to the authors have been answered and included in the current draft. The paper may be accepted. One suggestion to the authors: Please clarify what "payloads" mean in the context of a smart contract in Line 219.

Experimental design

-

Validity of the findings

-

Cite this review as